



# Characterization of OCO-2 and ACOS-GOSAT biases and errors for CO₂ flux estimates

Susan S. Kulawik[1], Sean Crowell[2], David Baker[3], Junjie Liu[4], Kathryn McKain[5,6], Colm Sweeney[5], Sebastien C. Biraud[7], Steve Wofsy[8], Christopher W. O'Dell[3], Paul O. Wennberg[9], Debra Wunch[10], Coleen M. Roehl[9], Nicholas M. Deutscher[11,12], Matthäus Kiel[4], David W.T. Griffith[11], Voltaire A. Velazco[11,13], Justus Notholt[12], Thorsten Warneke[12], Christof Petri[12], Martine De Mazière[14], Mahesh K. Sha[14], Ralf Sussmann[15], Markus Rettinger[16], Dave F. Pollard[17], Isamu Morino[18], Osamu Uchino[18], Frank Hase[19], Dietrich G. Feist[20,21,22], Sébastien Roche[10], Kimberly Strong[10], Rigel Kivi[23], Laura Iraci[24], Kei Shiomi[25], Manvendra K. Dubey[26], Eliezer Sepulveda[27], Omaira Elena Garcia Rodriguez[27], Yao Té[28], Pascal Jeseck[28], Pauli Heikkinen[23], Edward J. Dlugokencky[5], Michael R. Gunson[4], Annmarie Eldering[4], David Crisp[4], Brendan Fisher[4], Gregory B. Osterman[4]

[1] BAER Institute, 625 2nd Street, Suite 209, Petaluma, CA, USA

[2] GeoCarb Mission, University of Oklahoma, Norman, OK, USA

[3] Cooperative Institute for Research in the Atmosphere, Colorado State University, Fort Collins, CO, USA

[4] Jet Propulsion Laboratory, California Institute of Technology, Pasadena, CA, USA

[5] National Oceanic and Atmospheric Administration, Earth System Research Laboratory, Boulder, CO, USA

[6] University of Colorado, Cooperative Institute for Research in Environmental Sciences, Boulder, CO, USA

[7] Lawrence Berkeley National Laboratory, Earth Science Division, Berkeley, CA, USA

[8] Harvard University, Cambridge, MA, USA

[9] Division of Geological and Planetary Sciences, California Institute of Technology, Pasadena, CA, USA

[10] Department of Physics, University of Toronto, Toronto, Canada

[11] Centre for Atmospheric Chemistry, School of Earth, Atmosphere and Life Sciences, University of Wollongong, Wollongong, NSW, Australia

[12] University of Bremen, Bremen, Germany

[13] Oscar M. Lopez Center for Climate Change Adaptation and Disaster Risk Mgmt. Foundation Inc., Manila, Philippines

[14] Royal Belgian Institute for Space Aeronomy, Brussels, Belgium

[15] Karlsruhe Institute of Technology, IMK-IFU, Garmisch-Partenkirchen, Germany

[16] Karlsruhe Institute of Technology (KIT), Institute of Meteorology and Climate Research (IMK-IFU), Garmisch-Partenkirchen, Germany

[17] National Institute of Water and Atmospheric Research, Lauder, New Zealand

[18] National Institute for Environmental Studies (NIES), Tsukuba, Japan

[19] Karlsruhe Institute of Technology, IMK-ASF, Karlsruhe, Germany





[20] Lehrstuhl für Physik der Atmosphäre, Ludwig-Maximilians-Universität München, Munich, Germany

[21] Deutsches Zentrum für Luft und Raumfahrt (DLR), Institut für Physik der Atmosphäre, Oberpfaffenhofen, Germany

[22] Max Planck Institute for Biogeochemistry, Jena, Germany

[23] Finnish Meteorological Institute, Sodankylä, Finland

[24] NASA Ames Research Center, Moffett Field, CA, USA

[25] EORC Earth Observation Research Center, JAXA Japan Aerospace Exploration Agency

[26] Los Alamos National Laboratory, Los Alamos, NM, USA

[27] Izaña Atmospheric Research Center, Meteorological State Agency of Spain (AEMet), Tenerife, Spain

[28] LERMA-IPSL, Sorbonne Université, CNRS, Observatoire de Paris, Université PSL, Paris, France

*Correspondence to*: Susan S. Kulawik (susan.s.kulawik@nasa.gov)

**Abstract.** We characterize the magnitude of seasonally and spatially varying biases in the National Aeronautics and Space Administration (NASA) Orbiting Carbon Observatory-2 (OCO-2) Version 8 (v8) and the Atmospheric $CO_2$ Observations from Space (ACOS) Greenhouse Gas Observing SATellite (GOSAT) version 7.3 (v7.3) satellite $CO_2$ retrievals by comparisons to measurements collected by the Total Carbon Column Observing Network (TCCON), Atmospheric Tomography (ATom) experiment, and National Oceanic and Atmospheric Administration (NOAA) Earth System Research Laboratory (ESRL) and
U. S. Department of Energy (DOE) aircraft, and surface stations. Although the ACOS-GOSAT estimates of the column averaged carbon dioxide ($CO_2$) dry air mole fraction ($XCO_2$) have larger random errors than the OCO-2 $XCO_2$ estimates, and the space-based estimates over land have larger random errors than those over ocean, the systematic errors are similar across both satellites and surface types, $0.6 \pm 0.1$ ppm. We find similar estimates of systematic error whether dynamic versus geometric coincidences or ESRL/DOE aircraft versus TCCON are used for validation (over land), once validation and co-
location errors are accounted for. We also find that areas with sparse throughput of good quality data (due to quality flags and preprocessor selection) over land have ~double the error of regions of high-throughput of good quality data. We characterize both raw and bias-corrected results, finding that bias correction improves systematic errors by a factor of 2 for land observations and improves errors by ~0.2 ppm for ocean. We validate the lowermost tropospheric (LMT) product for OCO-2 and ACOS-GOSAT by comparison to aircraft and surface sites, finding systematic errors of ~1.1 ppm, while having 2-3 times
the variability of $XCO_2$. We characterize the time and distance scales of correlations for OCO-2 $XCO_2$ errors, and find error correlations on scales of 0.3 degrees, 5-10 degrees, and 60 days. We find comparable scale lengths for the bias correction term. Assimilation of the OCO-2 bias correction term is used to estimate flux errors resulting from OCO-2 seasonal biases, finding annual flux errors on the order of 0.3 and 0.4 PgC/yr for Transcom-3 ocean and land regions, respectively.



## 1 Introduction

The NASA OCO-2 instrument, launched in 2014, collects near infrared observations of reflected sunlight that are analyzed to yield estimates of the column-averaged $CO_2$ dry air mole fraction ($XCO_2$) (Crisp et al., 2015; Eldering et al., 2017; Crisp et al., 2017). OCO-2 is in sun-synchronous orbit with overpass about 1:36 pm local time. Soundings are collected continuously at 0.333 second intervals across a narrow swath (8 observations wide) while over the sunlit part of the globe. Adjacent ground tracks are separated by ~25 degrees of longitude (~2700+ km at the equator) on any given day, and repeated at 16-day intervals.

After each 16-day period, there are still gaps of about 1.6 degrees in longitude (e.g. see maps in Fig. 3). OCO-2 has 3 science observation modes: land nadir, land glint, and ocean glint. Nadir observations look straight down from the satellite and glint observations look slightly offset from the solar reflected glint spot (resulting in seasonal shifts in longitude for glint observations). OCO-2 additionally takes dense observations targeting TCCON stations or other sites. This targeted data has previously been validated for OCO-2 v7 (Wunch et al., 2017). This work focuses on validation of OCO-2 v8 with standard

mode observations. We validate both $XCO_2$ and the lowermost tropospheric column (LMT) introduced in Kulawik et al. (2016).

OCO-2 continues the $CO_2$ near infrared column record from GOSAT (launched in 2009), and from SCIAMACHY (launched in 2002). OCO-2, however, has better precision, a smaller footprint, and collects about a hundred times more observations than GOSAT. Use of $CO_2$ data for flux estimates is challenging because $CO_2$ is a long-lived atmospheric constituent, and

therefore variations in $CO_2$ are a small percent of the total amount of $CO_2$ (with typical seasonal and year-to-year variations between 0.1% and 3% of the $CO_2$ volume mixing ratio (VMR)). Similarly, because $CO_2$ is a long-lived atmospheric constituent, models are needed to trace the observed $CO_2$ anomalies back to surface sources or sinks. Models assimilate observations on the model grid, which typically has a resolution of 50 to 2500 km. Therefore, it is important to characterize how errors decrease with averaging, and to characterize both random and systematic errors, so that the data are assimilated

correctly. We also characterize two pre-averaged $XCO_2$ products.

For validation, this paper compares the satellite $XCO_2$ estimates to:

- $XCO_2$ derived from the Atmospheric Tomography (ATom) experiment (Wofsy et al., 2018), which provides validation snapshots at particular times over large latitude ranges;
- Columns from the Total Carbon Column Observing Network (TCCON) (Wunch et al., 2011, 2015) which provides
a time series of validation at specific locations;
- $XCO_2$ derived from aircraft observations by NOAA ESRL (Sweeney et al., 2015, 2018) and DOE (Biraud et al., 2013); and
- $CO_2$ from surface observations (Dlugokencky et al., 2018)

This paper attempts to characterize the following:

- OCO-2 and ACOS-GOSAT random error (for all observing modes) for $XCO_2$ and LMT;
- OCO-2 and ACOS-GOSAT systematic error (for all observing modes) for $XCO_2$ and LMT;
- Errors of pre-averaged OCO-2 products;
- Effects of bias correction;
- The size and correlation lengths of spatially and seasonally varying biases for OCO-2; and





•   Estimate of the impact of OCO-2 "regional" biases onto fluxes

In this paper, "systematic error" refers to the magnitude of seasonally and spatially varying biases, representing differences from validation data that do not reduce with averaging. In this work, two components make up the "systematic error" 1) "station bias variability", the standard deviation of the overall bias at each validation location, and 2) "correlated error", the standard deviation of n averaged satellite observations versus validation data, in the limit of large n. The correlated error

can result from aerosols or albedo, seasonally-dependent biases, or other conditions resulting in regional scale transient biases.

## 2 Datasets

### 2.1 Validation datasets

#### 2.1.1 ATom dataset

The ATom dataset is from an observation package containing 10-second averages, obspack_co2_1_ATom_v2.0_2017-07-10. This contains observations from the first and second ATom deployments. Like the HIAPER Pole-to-Pole Observations (HIPPO) observations used for $XCO_2$ validation in Frankenberg et al. (2016), the ATom flights collected in situ $CO_2$ profiles at altitudes extending from the surface up to altitudes between ~7 and ~13 km, with most flights going to 10 km. Figure 1 shows the ATom observation locations. The profiles are extrapolated upwards to the top of the atmosphere using model data,

as described later.

#### 2.1.2 ESRL and DOE aircraft datasets

Aircraft and ocean measurements taken by NOAA's ESRL and DOE (Biraud et al., 2013) are obtained from an observation package product (Sweeney et al., 2015, 2018). Because aircraft measurements are collected one or two times per month, and a tight geometric (+/-3 degrees latitude, +/- 5 degrees longitude, ±1 hour) coincidence criteria was adopted here, the validation

measurements are very sparse. In Kulawik et al. (2016), the geometric coincidence requirement was taken to be ±3 degrees latitude, ±5 degrees longitude, and ±1 week time, resulting in larger co-location error for geometric coincidences. Similarly as for ATom, the profile must be extended above the aircraft's maximum altitude using model fields, and the profile is extended to the surface using the lowest observation (between 0.2 and 0.5 km above the surface).

| The aircraft stations used for land validation are: | The aircraft stations used for ocean validation are: |
|---|---|
| North Slope, Alaska (NSA) | Estevan Point, Canada (ESP) |
| Poker Flat, Alaska (PFA) | Offshore Portsmouth, New Hampshire (NHA) |
| East Trout Lake, Canada (ETL) | Trinidad Head, California (THD) |
| Dahlen, North Dakota (DND) | CMA, Offshore Charleston, South Carolina (SCA) |





Park Falls, Wisconsin (LEF)

Offshore Portsmouth, New Hampshire (NHA)

Trinidad Head, California (THD)

Briggsdale, Colorado (CAR)

Homer, Illinois (HIL)

Offshore Cape May, New Jersey (CMA)

Southern Great Plains, Oklahoma (SGP)

Offshore Charleston, South Carolina (SCA), and

Offshore Corpus Christi, Texas (TGC)

Offshore Corpus Christi, Texas (TGC), and Rarotonga,

Cook Islands (RTA)

(Note that RTA is the only remote ocean site)

The version of ESRL aircraft data (Sweeney et al. 2018) analyzed for this paper has since been updated to exclude ~5% of $CO_2$ measurements due to a negative bias related to ambient $H_2O$ levels greater than ~1.7 % v/v. We investigated the potential impact of the biased data using a candidate correction factor applied to data from the RTA site, where the largest fraction of data were affected, and found maximum potential errors of 0.1 ppm in $XCO_2$ and 0.4 ppm in LMT. This error is captured in Table 1 and subtracted from the second-to-last column in Tables 2-5 to estimate the satellite error. Since the LMT overall bias

is tied to the aircraft (similarly to how $XCO_2$ is tied to TCCON), this may result in an overall bias in LMT.

### 2.1.3 The TCCON dataset

The GGG2014 TCCON dataset is used in this paper (Wunch et al., 2015). The locations of the TCCON stations used for this work are shown in Figure 1b. The TCCON data are averaged over 90 minutes to reduce error. It would be more ideal to average TCCON data ± 60 or 90 minutes within each satellite observation, rather than pre-averaging the TCCON data, but due

to processing time, the pre-averaging approach was used. As previously discussed, OCO-2 has different observing modes, and different TCCON sites can be used for the validation of one or more mode types. The TCCON $XCO_2$ estimates are also derived from remote sensing observation, though with extensive aircraft validation to tie them to WMO standards (Wunch et al., 2011). They typically have a systematic error of about 0.4 ppm for averaged values (Wunch et al., 2017), which is captured in Table 1 as "TCCON error". Because the TCCON $XCO_2$ estimates are derived from up-looking measurements of direct

sunlight, they have a somewhat different vertical sensitivity to $CO_2$ than OCO-2 or GOSAT $XCO_2$. The TCCON averaging kernel varies with solar zenith angle and altitude, and 18 column averaging kernels are provided by the TCCON team at increments of 5 degrees from 0-85 degrees. The TCCON averaging kernel is applied:

$$\mathbf{x}_{ret} = \mathbf{x}_{prior} + \mathbf{A}(\mathbf{x}_{true} - \mathbf{x}_{prior}) \qquad\qquad (1)$$

where $\mathbf{x}_{true}$ is the "true state", $\mathbf{x}_{prior}$ is the prior vector, and $\mathbf{x}_{ret}$ is the simulated retrieved value. For our case, $\mathbf{x}_{true}$ is the

retrieved TCCON profile, which is the TCCON prior scaled to the retrieved value, and $\mathbf{x}_{ret}$ is the TCCON retrieval with OCO-2 averaging kernel applied. Note that TCCON and ACOS-GOSAT and OCO-2 all use the same prior vector, which assumes





a consistent yearly increase that has been lagging behind from the actual $XCO_2$ increase over the past several years and will be updated in the next TCCON version.

Two stations have special considerations regarding altitude: The Izaña station is on an island (mean surface pressure 773 hPa),

surrounded by ocean (mean surface pressure 1016 hPa); Garmisch is in a local valley (mean surface pressure 840 hPa) surrounded by higher elevation land (though the mean pressure of OCO-2 co-located within 3 degrees latitude and 5 degrees longitude is 987 hPa). For both of these cases, the $XCO_2$ was calculated using the pressure grid of the original observation. The station-dependent results (in Tables B.1 and B.2) show that these two stations without any additional manipulations to match the pressure grids, have results similar to other TCCON stations and so we did not try to match the pressure grids. A

more comprehensive discussion of the effects of surface pressure on validation is in Kulawik et al. (2016). It would also make sense to use a more relaxed (±2 hours) co-location for geometric coincidence at remote ocean sites (e.g. Reunion, Ascension), but we found sufficient co-locations with ±1 hour and so used consistent criteria at all sites.

TCCON stations used in this analysis for land coincidences are: Eureka (Strong et al., 2017), Sodankyla (Kivi et al., 2017; Kivi and Heikkinen, 2016), East Trout Lake (OCO-2 only, Wunch et al., 2017), Bialystok (Deutscher et al., 2017), Karlsruhe

(Hase et al., 2017), Orleans (Warneke et al., 2017), Garmisch (Sussmann and Rettinger, 2014), Park Falls (Wennberg et al., 2017), Lamont (Wennberg et al., 2017), Anmyeondo (Goo et al., 2017), Dryden (Iraci et al., 2017), Hefei (Liu et al., 2018), Burgos (OCO-2 only, Velazco et al., 2017), Manaus (only land glint OCO-2, Dubey et al., 2017), Darwin (Griffith et al., 2017), Wollongong (Griffith et al., 2017), Lauder125 (Sherlock et al., 2017).

TCCON ocean stations used in this analysis are: Park Falls (OCO-2 glint, over the Great Lakes, Wennberg et al., 2017),

Rikubetsu (OCO-2, Morino et al., 2017), Tsukuba125 (Morino et al., 2017), Saga (Shiomi et al., 2017), Izana (Blumenstock et al., 2017), Ascension (Feist et al., 2017), Darwin (Griffith et al., 2017), Reunion (De Mazière et al., 2017), Wollongong (Griffith et al., 2017), Lauder125 (Sherlock et al., 2017).

Stations not used in calculating error fits, due to local effects are: JPL, Caltech, FourCorners, Indianapolis, Paris, and Bremen. Tsukuba has some influence from nearby Tokyo but was included in the analysis. Other stations not included were due to

insufficient co-locations (<100) (Ny Alesund, Zugspitze).

### 2.1.4 Surface flask measurements

Remote surface sites are from the ESRL Observation Package Data Product surface flask measurements (Conway et al., 1994). These           were           downloaded           on           9/2018           from https://www.esrl.noaa.gov/gmd/ccgg/obspack/data.php?id=obspack_co2_1_GLOBALVIEWplus_v4.0_2018-08-02

(Dlugokencky et al., 2018) and include local times between 11:30 am and 3:30 pm. The "remote oceanic" locations used in this paper are selected to have at least 97% ocean along a circle with a 5 degree radius around the location, which are summarized here and shown in Fig. 1:

- ena at 39°N, 28°W
- mid at 28°N, 177°W





- mnm at 24°N, 154°E
- mlo at 20°N, 155°W
- asc at 8°S, 14°W
- smo at 14°S, 171°W
- ams at 38°S, 78°

## 2.2 OCO-2 version 8 (9/2014 through 2/2018)

We validate the NASA OCO-2 v8 dataset using the bias-corrected Lite files, available from https://disc.gsfc.nasa.gov/datasets/OCO2_L2_Lite_FP_V8r/summary?keywords=OCO-2 (O'Dell et al., 2018). The documentation for this dataset is available from https://co2.jpl.nasa.gov/#mission=OCO-2. The algorithmic improvements in v8 include (1) using lamp, solar, and lunar data to produce a radiometric correction that accounts for the effects of icing, solar diffuser and lamp degradation, (2) spectroscopy updates, and (3) addition of a retrieved stratospheric aerosol, which improved biases seen near the southern and northern edges of OCO-2 data seasonally (described in Wunch et al., 2017 and O'Dell et al., 2018). The lower tropospheric (LMT) product for OCO-2 was calculated similarly to Kulawik et al. (2017) with bias correction described in Appendix A.

### 2.2.1 OCO-2 Averaged products

Averaging is useful for combining similar information over scales smaller than model resolution, to manage errors, and reduce dataset size. Averaging allows combining geographically similar observations and assigning an error to these assuming correlated and random error values. Assimilation methods generally assume errors from different observations are not correlated, and therefore assimilation of 1000 separate observations, each with error of 1 ppm, into a single grid box would result in errors of < 0.1 ppm, which is much smaller than the actual errors after averaging. Different averaging schemes are possible, e.g. weighting by predicted errors, checking agreement with neighboring observations (buddy check), and weighting by local variability. Three of these schemes are described below and validated against TCCON and aircraft data.

Averaged products can have a significantly reduced data volume. The number of good quality land glint observations co-located with Lamont TCCON (geometric co-location, ± 3 degrees latitude, ±5 degrees longitude, ± 1 hour) is:

- 175,000 for the OCO-2 lite product
- 1,300 for OCO-2 Baker-averaged product (described in Section 2.2.2)
- 1,200 for OCO-2 Liu-averaged product (described in Section 2.2.3)
- 8,700 for OCO-2 20-average product (described in Section 2.2.4)

For comparison, there are 7,000 quality observations for GOSAT nadir land (over 7 years).



### 2.2.2 v8 Baker "10-second" average

A 10-second average product (~0.6 degree) was created by David Baker (described in Crowell et al., 2019) to reduce the computational burden of assimilation and to assign errors to averaged quantities. For most assimilation systems, 0.6 degree is below the assimilation system resolution. This product averages 10 seconds of OCO-2 observations, spanning 0.6 degrees of latitude. If all soundings are cloud free and have converged, their quality flags are good, and a 10-second collection includes 240 observations. Because soundings that are cloudy or have bad quality flags are not included, the mean number of observations included in this averaged product is 74, and the number of observations per bin ranges from 1 to 240. This dataset emphasizes evenly weighting data at one-second intervals (Crowell et al., 2019), so averaging is first done for each one second of data (spanning ~0.06 degrees), weighting the average by the predicted total error from the v8 lite product. The error for the one-second average is the larger of the standard deviation of the data that is averaged or the mean predicted total error from the v8 lite product. Then, the one-second bins are averaged.

In the observations used in Table 5, there is an average of 85 observations per bin, 6% of the bins have 1 observation (representing 0.04% of the original data), 3.5% of the bins have 2 observations, 31% of bins have 19 or less observations per bin, 55% of bins have 74 or less, and 13% have 200 or more observations per bin. Eliminating bins with fewer than 20 soundings removes 2% of observations and 31% of the bins, so the averaged product has many bins that represent only a few measurements. We evaluate the Baker-10 second average product for all bins together, and for a subset of this product with bins containing at least 20 observations. The predicted error for this product decreases with the number of observations per bin, with an average error of 0.8 ppm when n=1, and an average error of 0.5 for n = 240.

### 2.2.3 v8 Liu "Buddy" average

The v8 Liu average product was generated in the same manner as the v7 Liu average product used in Liu et al., (2017). Only observations with "good quality" as indicated with the quality control flag were used. We use a two-step strategy to homogenize the observation spatial distribution and remove the outliers. We first perform a buddy-check (Liu et al., 2012) to remove the outliers within 100 km along the same orbit track. The observation, $y_i$, is removed if

$$\left| y_i - \frac{1}{n} \sum_{ii \in A} y_{ii} \right| > 2\sigma^o$$

where A represents 100 km circular domain, n is the total number of observations within the domain, and ii is within the domain. The value, $\sigma^o$, is the standard deviation of all the observations within the domain. We then calculate a super observation (or "superobs") for every 1° by 1° box by averaging the observations within the 1° by 1° domain at the same orbit track. Note that the 1° by 1° box does not have overlaps. The mean latitude, longitude, and time are the longitude, latitude, and time of the new average product.



The observation error of the average product is the sum of 1) the mean of the observation uncertainty from OCO-2 Lite files and 2) the standard deviation of the observations within the 1° by 1° box along the same orbit track. The means of the observation uncertainty within the 1° by 1° box are used to represent biases, assuming biases are dominant relative to random errors. The field "xco2_std+uncertainty" is the predicted error for the averaged product.

### 2.2.4 20-observation average

This average product is generated by looping over the OCO-2 observations ordered in time, and averaging 20 adjacent good quality observations, provided the observations are collected within 30 seconds and 1 degree of latitude of each other. Variations on this product were tested, e.g. weighting by the inverse predicted error or local variability. However, the more complicated averaging did not yield improved results, so unweighted averaging was used.

### 2.3 ACOS-GOSAT v7.3 (4/2009 through 5/2016)

The ACOS-GOSAT v7.3 product is an update from the previous v3.5, using a Level 2 (L2) retrieval algorithm that approximately matches that used to produce the OCO-2 v7 product (O'Dell et al., 2018). The key changes to the L2 algorithm relative to OCO-2 included scaling the $O_2$-A band spectroscopy, and fitting a zero level offset to the A-band (O'Dell et al., 2018). The v7.3 product includes observations collected from April, 2009 through May, 2016, which provides some overlap with OCO-2. The lower tropospheric (LMT) product for V7.3 was calculated similarly to Kulawik et al. (2017) with bias

correction described in Appendix A.

### 2.4 CarbonTracker 2017 (2009-2016)

The CarbonTracker model (Peters et al., 2007) output, CT2017, was downloaded from (ftp://aftp.cmdl.noaa.gov/products/carbontracker/co2/CT2017/molefractions/co2_total/) and goes through 12/31/2016. From 2017 onwards, the CarbonTracker model is extrapolated from 2016 by adding a global offset of 2 ppm/year. The

CarbonTracker model is used to quantify co-location error (by calculating biases and errors for the model co-located with validation locations and times versus satellite location and times). The CarbonTracker model is also used to extend aircraft profiles from the top observation to the top of the atmosphere. The CarbonTracker model through 2016 is also directly compared to TCCON and aircraft observations. Within the domain (22.5-61.5°N, 63.5-128.5°W), the 1x1 degree "nam" files are used, and elsewhere, the 3x2 degrees files are used.

### 265 2.5 CAMS Reanalysis (2009 through 2017)

The CAMS reanalysis model output was downloaded from http://apps.ecmwf.int/datasets/data/cams-ghg-inversions/ and goes through December 2017 (see description https://confluence.ecmwf.int/pages/viewpage.action?pageId=58131166). This model is used to extend aircraft profiles to the top of the atmosphere (up to 0.1 hPa, or 65 km) for evaluation of the error introduced



by model extension. Comparisons between profiles extended by different methods are used to quantify the error from
extending aircraft profiles.

### 2.6 GMAO-7km and GMAO-50km (6/2005 through 6/2007)

The Goddard Modeling and Assimilation Office (GMAO) produced a high resolution, global nature run for $CO_2$ (Putman et
al., 2014). This data is run/sampled at 7 km and 50 km gridding every 30 minutes. The run extends through half of 2005,
2006, and half of 2007. The difference between this model co-located to the validation data versus co-located to the satellite
observations is used to quantify co-location error. Everything is matched except the year, which is set to 2006. We are using
this to quantify the differences between $XCO_2$ at the satellite and validation locations and time. We were interested in whether
the smaller domain models would result in higher co-location errors.

### 3 Methods and quantification of imperfect validation

This section discusses co-location criteria, methodology used to estimate errors, and a quantification of the errors from
imperfect validation, which includes errors in the validation data itself (quantified by the data provider), errors from missing
validation data, errors resulting from sensitivity mismatches, and errors from imperfect co-location. Section 3.1 describes the
types and methodology used to estimate systematic and random errors.

### 3.1 Co-location criteria and examples of co-locations versus time and latitude

Two co-location criteria are used, geometric and dynamic (following the methodology of Kulawik et al. (2016), and similarly
to other OCO-2 and GOSAT validation criteria). Geometric is within 3 degrees latitude and 5 degrees longitude. For ATom
and ESRL/DOE aircraft, the time coincidence is 9 hours (same-day coincidence), and for TCCON the time coincidence is 1
hour. The dynamical criteria (Wunch et al., 2011; Keppel-Aleks et al., 2011; Keppel-Aleks et al., 2012) are designed to exploit
information about the dynamical origin of an air parcel through a constraint on the free-tropospheric temperature. Briefly, a
co-location is found when the measurements are within 5 days and the following is satisfied:

$$\left( \left( \frac{\Delta Latitude}{10} \right)^2 + \left( \frac{\Delta Longitude}{30} \right)^2 + \left( \frac{\Delta Temperature}{2} \right)^2 \right) < 1 \qquad (2)$$

"dynamic small" uses Eq. 2 with coincidence changed from 10 degrees to 5 degrees in latitude, 30 degrees to 15 degrees in
longitude, while still using 2 degrees K. Data are selected within $\pm$ 2.5 days rather than $\pm$5 days. "Dynamic small" has smaller
co-location error, but is not adequate for comparisons to TCCON stations north of 50N where OCO-2 does not reach in winter
months, whereas the standard TCCON co-location of $\pm$ 10 degrees latitude reaches more stations. Other co-location techniques
include Nguyen et al. (2014) which used spatiotemporal averaging, and Guerlet et al. (2013) which considered model gradients





when selecting co-locations. Since this work finds that co-location errors are significant, the more specialized co-location techniques of Nguyen et al. (2014) and Guerlet et al. (2013) will be helpful for future work.

Figures 2 and 3 show examples of geometric and dynamic co-location at different stations for OCO-2 and GOSAT. Use of the dynamic criteria results in about a thousand times more co-locations. In the lower panel of Fig. 3b, a bias between OCO-
2 and TCCON in the summer, 2016 is seen for dynamic co-location. This bias is not seen for geometric co-location (lower panel of Fig. 3a). The co-location error is quantified by the difference between the CarbonTracker model co-located with the satellite and with the TCCON observations, shown by the purple or black dashed line in the difference plots. This model/model comparison compares the model fields versus model fields at different times and locations. The co-location error indicates a bias is expected for OCO-2 minus TCCON in summer, 2016, though it is a smaller bias than the actual bias of OCO-2 minus
TCCON.

As seen in Figure 2, dynamic criteria are useful for GOSAT which does not have sufficient co-locations at monthly 3x5 degree bins at many stations for geometric co-location. Although OCO-2 collects an order of magnitude more data than GOSAT, dynamic coincidence criteria are still useful for OCO-2 because high latitudes do not have data in winter months. Figure 4 shows co-locations at Park Falls (45.945N). Poleward of 40N has no co-located OCO-2 observations in winter due to quality
flag screening during winter. However, dynamic coincidence fills in these temporal gaps.

Figure 5 shows comparisons of OCO-2 $XCO_2$ and lower partial column (LMT) versus DOE aircraft observations at sgp (36.6N, 97.5W). The aircraft profile is extended with CarbonTracker2017 and the OCO-2 averaging kernel is applied using Eq. 1. Note that the lower partial column shows about twice the seasonal cycle amplitude as the full column, and also about twice the error. The dashed line in the lower panels shows the difference between the model at the satellite observation locations
and times minus the model at validation locations and times. The co-location flags a large co-location error in mid-2015, where the satellite minus aircraft shows a similar error. Although the patterns look similar, one seasonal minimum and maximum for $XCO_2$ in 2015 are shown on the both graphs. The LMT reaches the maxima and minima at least one month before $XCO_2$.

Figure 6 shows OCO-2 versus ATom aircraft observations (where the aircraft observations are extended using CarbonTracker
model and the OCO-2 averaging kernel is applied, using Eq. 1) using geometric coincidence criteria. The gaps are locations where no coincidences are found with good quality OCO-2. An overall low bias is seen for the "1S" campaign, but much less bias is seen for the "2N". This could indicate a temporal (seasonal) bias, which is discussed later in the paper.

**3.2 Station bias and bias variability: grouping comparisons versus validation data to determine regional biases**

The TCCON dataset provides a timeseries of validation data at particular locations; whereas the ATom dataset provides a
range of latitudes at particular times. The validation data are grouped by "station", which is a TCCON location, an ESRL location, or an ATom campaign. The overall bias for each station is called the "station bias" and gives an idea of the sizes of regional biases. Table 2 shows the station bias for each coincidence type. The only category in Table 2 that had an overall bias was ATom versus OCO-2 ocean glint, which had a -0.7 ppm bias (OCO-2 low). Appendix B shows biases for each





individual station and campaign for OCO-2. For land, in Table B.1, these biases do not seem to show any consistent pattern,
however, for ocean, seen in Table B.2, OCO-2 versus the ocean stations between 19 and 33N (Burgos, Izana, and Saga) have
biases from -0.7 to -1.0 ppm. The low ATom bias seen in Table 2, however, is persistent over all latitudes (e.g. see OCO-2
versus ATom Fig. 6a). However, note that the ATom Atlantic track (2N) does not have a low bias, except around 10-30N, as
seen in Fig. 6b, consistent with the low bias seen at Izana. A persistent bias can be due to the particular types of aerosols or
albedos near a station, a time-dependent bias, or other persistent factors. So, in summary there seems to be no trend in land
biases, but there seems to be persistent low biases in OCO-2 ocean glint observations, at least between 19 and 33N.

The variability of the station bias is characterized as the "station bias variability" and feeds into the systematic error estimate.
For example, the average difference (mean bias) of all co-location pairs of (satellite – TCCON) for the set of stations north of
50N (Eureka, Sodankyla, East Trout Lake, and Bialystok) are -1.5 ppm, 0.6, -0.2 ppm, and 0.3, respectively. The "station bias
variability" for the stations north of 50N is standard deviation of these mean biases, or 0.9 ppm. The station bias variability is
calculated for all stations and is part of the systematic error (the total systematic error combines the station bias variability and
the correlated error discussed in Section 3.3 and is calculated in Section 4.2).

### 3.3 Quantifying how error reduces with averaging

OCO-2 collects stripes of densely packed data, with ~200 observations for 0.5 degrees latitude. When OCO-2 data is
assimilated into models with resolution of 0.5 degrees or more, hundreds to thousands of observations can be averaged and
compared to a particular model location and time. It is important to characterize the error of the averaged quantity. Kulawik
et al. (2016) characterized the correlated and random errors of ACOS-GOSAT and SCIAMACHY observations, finding that
satellite errors reduce when $n$ observations are averaged according to Equation 3 (where all $n$ observations are co-located with
the same validation data point). For ACOS-GOSAT v3.5, the correlated error was about 0.8 ppm and the random error about
1.6 ppm. Worden et al. (2017) also found that OCO-2 v7 data had correlated errors within 1-degree bins.

$$error = \sqrt{correlated^2 + random^2/n} \qquad\qquad (3)$$

The correlated error is a systematic error and is combined with the station bias variability to estimate the total systematic error.

### 3.4 Quantification of uncertainty introduced from the validation process

This section quantifies the uncertainty introduced in the validation process, for example, the error resulting from a spatio-
temporal mismatch of validation and satellite data or aircraft profile extension. This section describes and quantifies validation
errors, which in total give an upper bound on the validation accuracy. These errors quantified result from:

- Co-location error. Validation data rarely occurs exactly at the point of satellite observation; and even if exactly
coincident, does not integrate over the entire airmass that the satellite observes (e.g. 2x3 km footprint). Co-location
error is quantified by comparing the model co-located to the observation time/location of the satellite versus the
model co-located to the observation time/location of the validation data. The comparison of model@satellite vs.
model@validation is used to compute co-location station biases, co-location correlated errors, and co-location
random errors. These co-location error estimates are shown in Table 1 and Table 2. Table 1 shows an estimate for


different coincidence criteria and viewing modes. Table 2 shows the co-location error for each entry (shown in parentheses) and then subtracts this error in quadrature to estimate a co-location corrected systematic error. 4 different model runs are used to estimate co-location error: CT2017, CAMS, GMAO50, and GMAO7, with GMAO7 showing the highest co-location errors, of 0.5 ppm, indicating higher variability in GMAO7. However, CT2017 is used in Table 2 because it has results for all years. The co-location error in Table 1 was for OCO-2 comparisons to TCCON and gives a general idea of the magnitude. In Table 2, the co-location error was estimated for every entry individually.

- Averaging kernel error (TCCON). This is due to differences between the TCCON and OCO-2 sensitivity to the true $CO_2$ profile and estimated 1) using comparisons with and without the OCO-2 averaging kernel applied to TCCON, and 2) applying Eq. 1 to CarbonTracker model profiles to simulate TCCON (sim_TCCON) or OCO-2 results (sim_OCO2), and then look at the difference between the sim_OCO2 and sim_TCCON, which tests the effects of the different sensitivity of OCO-2 and TCCON. For 1), comparing OCO-2 v8 vs. TCCON found that systematic errors decreased from 0.58 to 0.50 ppm for ocean and from 0.82 to 0.72 ppm for land if the OCO-2 averaging kernel is applied to TCCON prior to comparison to OCO-2 (using Eq. 1). Test 2) was done at Park Falls and Darwin, finding about a seasonally dependent 0.2 ppm amplitude difference at Park Falls (sim_OCO2 higher in the summer), and ~0.1 ppm seasonally dependent difference at Darwin (sim_OCO2 higher in the months SON). The difference also has a slope of -0.1 ppm/year (sim_OCO2 increasingly lower) due to the increasingly discrepancy between the TCCON/ACOS prior and CarbonTracker (e.g. see Fig. 3). The difference between sim_TCCON with the OCO-2 averaging kernel applied and sim_OCO2 was tested, but the results were the same. Previously, Wunch et al. (2011, appendix A) estimated this size as ~0.2 ppm and Nguyen et al. (2014) estimated this as 0.12 ppm for ACOS-GOSAT. This error is set to 0.1 ppm in Table 1, but unlike other errors it is directly subtracted, not subtracted in quadrature, because the comparison with and without averaging kernel resulted in a similar error difference no matter the size of the total error (e.g. 0.4->0.3, or 0.8->0.7 ppm). Note that the TCCON averaging kernel was not applied to OCO-2 because that process needs the retrieved OCO-2 profile and it was not clear how to bias correct the OCO-2 profile. Kulawik et al. (2017) showed that the bias correction is significantly different in the near surface versus tropospheric part of the profile.
- Aircraft profile extension to the top of the atmosphere (TOA). Uncertainties introduced by aircraft profile extension above the aircraft observations. An aircraft profile is never measured over the entire range of the surface to the top of the atmosphere. Model data is used to extend the profile over the whole atmosphere (up to 65 km). Errors resulting from aircraft profile extension are estimated by comparing different extension schemes, either from using different models or methodologies (e.g. straight extension vs. scaled extension), described below (although it is possible that different models or schemes have biases in the same direction).
- Aircraft profile sampling and extension to surface. Uncertainties from aircraft extension to the surface. The lowest aircraft measurement is 0.2 to 0.5 km. This was estimated using AirCore profiles and sampling at aircraft altitudes for SGP, then calculating the total column either with the original AirCore profile or the sampled profile, described in more detail below.

The errors for each of the above are shown in Table 1.

This paragraph discusses the aircraft profile extension error to the top of the atmosphere (TOA). ATom profiles go up through about 9 or 13 km, whereas ESRL and DOE aircraft profiles go up through 3 to 9 km. The aircraft profile is extended in 4 different ways using two independent models; then the differences in validation give an estimate of the error introduced by profile extension. The different ways the profile is extended are 1) "tropo": extending the top value of the aircraft profile up through the NCEP identified tropopause, then tacking on the CarbonTracker model, 2) "scale": calculating an offset between the CarbonTracker model and the top observed aircraft observation, then extending the profile upwards using the





CarbonTracker model plus offset; 3) and 4) Same as 1) and 2) but using the CAMS model. For ESRL land, there were no statistical difference in any aspect of the comparison, but there was an overall bias difference of 0.1 ppm (CarbonTracker results are higher by 0.1 ppm). ESRL "ocean" comparisons use stations at land/ocean boundaries, e.g. LEF near the Great Lakes, PFA on the coast of Alaska, TGC at the border of Texas and the Gulf of Mexico. For these comparisons, the LEF site showed large differences (with the "scale" extension resulting in lower bias variability for OCO-2 versus aircraft among all

aircraft sites). The LEF site is unique in that it only goes up to 3 km, whereas other sites go up to 5-9 km. The LEF site did not show differences for extension with CAMS versus CarbonTracker. When the LEF site was removed, there were small differences (< 0.1 ppm) between "tropo" and "scale", and no differences between using the CarbonTracker versus CAMS models for extension. For the ATom profiles, extension with CarbonTracker also resulted in a -0.1 ppm bias versus extension with CAMS (the opposite sign as the above). However, there were no other differences in comparison metrics. So aircraft

profile extension was estimated as 0.1 ppm. This is smaller than the previous calculation of this in Wunch et al. (2010) of 0.3 ppm. Wunch et al. (2010) perturbed the model profile to check the size of the effects, whereas this work compared results when using 4 different but equally plausible methods of profile extension.

This paragraph discusses errors introduced by aircraft sampling and extension to the surface. This analysis was done for ESRL aircraft sampling at the SGP site. The sampling was at 0.50, 0.63, 0.94, 1.25, 1.58, 1.92, 2.25, 3.19, 3.84, 4.19, 4.83 km above

sea level (ASL). The surface at SGP is at 0.31 km above sea level. 3 AirCore profiles taken at SGP were degraded to aircraft sampling, then $XCO_2$ was calculated from the sampled profiles and compared to the original AirCore $XCO_2$. The differences for the above were less than 0.1 ppm. The AirCore profile varies less than 1 ppm between the surface and the lowest aircraft measurement. The air between the surface and 0.2 ppm above the surface represents 2% of the total airmass, so that this could be used to estimate the error from extending the aircraft measurement to the surface.

The total systematic error added from validating from $XCO_2$ constructed from aircraft profiles is estimated as ~0.2 ppm, from Table 1. This combines errors from the aircraft data (Section 2.1.2) and errors from the profile extension at the top of the atmosphere and at the surface. This error estimate is less than the ~1 ppm estimated in Miyamoto et al. (2013), which was estimated at airports where extension by models near the surface will be much less accurate.

**4 Random and systematic errors**

This section quantifies random and systematic errors for ACOS-GOSAT and OCO-2, and also for CarbonTracker. One important component of this process is the quantification and removal of validation error, which is discussed in section 3.4 and Table 1.

**4.1 Plots showing how error reduces with averaging**

After satellite data is co-located to validation data (described in Section 3.1), the overall bias is subtracted (described in section

3.2), and the standard deviation of the satellite data minus validation data is calculated. How the standard deviation of satellite





data minus validation data decreases as more satellite data is averaged is quantified. Figure 7 shows how error reduces when data is averaged. As more data is averaged, the error is reduced. However the error does not drop off as the inverse of the square root of n, where n is the number of data points averaged, but rather as Eq. 3, which has both a correlated and random component.

## 4.2 Characterization of random and systematic errors


This section quantifies random errors (errors which improve with averaging, see Section 4.1) and systematic errors, composed of regional biases (station bias variability, see Section 3.2) and correlated errors (errors that do not improve with averaging (see Section 4.1)). The correlated error and station bias variability are added in quadrature to estimate the total systematic error. The results of these, by satellite, viewing mode, and coincidence criteria are shown in Table 2. The values in "()" are

estimates of the co-location errors, which are obtained from the difference between the model co-located to validation data and model co-located to the satellite (discussed in Section 3.4). The column, "Co-location corrected systematic error" subtracts out the co-location error in quadrature. Dynamic co-locations have, in general, higher systematic error and higher co-location error; however, the corrected systematic error is fairly consistent between geometric and dynamic criteria. The final column subtracts other validation errors, which are TCCON error, Averaging Kernel difference error (for TCCON), Aircraft error,

uncertainty from Aircraft profile extension to TOA, and Aircraft profile sampling and extension to surface, to estimate the systematic error for the satellite.

The OCO-2 land glint and land station bias variability without Manaus and Hefei is 0.4 ppm rather than 0.6, due to the large mean biases at Manaus and Hefei of -1.6 and -1.2 ppm, respectively (OCO-2 lower than TCCON). The model-predicted co-location bias at Manaus and Hefei are 0.1 and 0.3 ppm, respectively, so do not explain the large biases (which could reflect

satellite errors or models failing to capture the gradients at these locations). Without these two stations, the station bias variability, subtracting the co-location error in quadrature, is 0.4 for all modes of GOSAT and OCO-2.

The total systematic error is fairly consistent at 0.5 – 0.8 ppm for ACOS-GOSAT and OCO-2 land. The lower values for geometric coincidence for ocean OCO-2 observations of 0.3 – 0.4 ppm should also consider the overall bias of OCO-2 ocean, of -0.2 to -0.7, and the estimate of 0.6 ppm for OCO-2 ocean dynamic coincidences. The comparisons of ESRL aircraft versus

OCO-2 land observations give consistent results to TCCON, but comparisons to OCO-2 ocean observations estimates a higher systematic error than TCCON, possibly because of onshore/offshore differences not captured in the co-location error.

ACOS-GOSAT has about 1.6 times higher random error than OCO-2 but similar total systematic error to OCO-2. The CarbonTracker model also has somewhat smaller systematic errors than the satellite data, estimated at about 0.4 ppm.

In Table 3, we show errors on the raw (non-bias corrected) data. The bias correction improves the overall bias, the systematic

error (by about 0.2 ppm for ocean and about 0.5 ppm for land), and improves the random error by about 0.5 ppm larger for OCO-2 land and GOSAT ocean and improves OCO-2 random error by 0.2 ppm over ocean.

Table 4 shows validation of lower tropospheric (LMT) OCO-2 and GOSAT described in Kulawik et al. (2017). We cannot compare versus TCCON because TCCON is a total column measurement, so LMT is compared to aircraft and remote surface





site observations. The OCO-2 and ACOS-GOSAT land LMT systematic errors are 0.8 and 1.1 ppm, respectively. The ocean
validation relies on OCO-2 ocean observations near continental aircraft sites. The systematic errors for ocean LMT for OCO-2 and ACOS-GOSAT are higher than expected, at 3.0 and 1.6 ppm, respectively. However, note that the estimate for ocean $XCO_2$ versus aircraft shown in Table 2 is significantly higher than the TCCON or ATom error estimates for ocean, possibly because of on-shore/off-shore differences. Also, note that the one true oceanic site, RTA, validates well, with ~1.0 ppm systematic error. The OCO-2 versus ATom comparisons show a systematic error of <1.1 ppm for ocean comparisons. The
actual error cannot be determined because the co-location error is also 1.1 ppm and when it is subtracted the predicted error is 0. Satellite observations are also compared to surface sites, with ~1.3 ppm systematic error. The surface sites are at the surface, rather than the lowermost column from satellites, so this estimate is likely high. Considering all of the above, the systematic error for OCO-2 and ACOS-GOSAT is estimated conservatively at ~1.1 ppm. However, recently aircraft measurements were found to have a low bias on the order of -0.6 ppm for LMT and that the LMT overall bias is tied to aircraft.
So the v8 product may have a -0.6 ppm low bias. This is consistent with the low biases versus ATom (-0.8 ppm) and surface observations (-0.3 ppm) in Table 4.

Although 1.1 ppm is higher than the $XCO_2$ systematic error of ~0.6, the true variability of LMT is also larger than $XCO_2$. Looking at a few random days: 6/16/2015, 10/16/2015, 1/16/2015, the standard deviation of the values of LMT divided by the variability of $XCO_2$ was 2.1, 2.4, and 3.4, respectively, and the correlation between LMT and $XCO_2$ was 0.71, 0.81, and
0.90, respectively. So LMT has 2-3 times the signal variability of $XCO_2$. Figure 5 shows time series of $XCO_2$ and LMT versus aircraft observations. This figure shows both the higher variability of LMT and the offset seasonal cycle between $XCO_2$ and LMT.

**4.3 Results for averaged data products**

The pre-averaged products are useful for assimilation, as the data volume is significantly less, and the averaged product has
an assigned error for each averaged value. We calculate the standard deviation of the averaged products versus validation data. Since the data is already significantly averaged, the characterization of how the error reduces with averaging is more difficult to quantify with the averaged products.

Table 5 shows results for averaged data products. The Baker product has an average of 74 observations per bin. The expected standard deviation of 74 average observations for land nadir, using Eq. 3 and Table 2, would be 0.8 ppm, whereas the Baker
product standard deviation is 1.3 ppm. However, even though the average number of observations per bin is 74, the range of observations per bin is 1 to 240, and the averages consisting of a lower number of observations will have higher errors. For this reason, Table 5 also shows the standard deviation of the Baker product only for bins when the number of observations per bin is at least 20. The standard deviation of this product is 0.9 ppm, which is similar to the predicted error and the same as the straight average 20-obs average of the lite product. The Liu product for v8 also has a higher standard deviation than would be
expected for an averaged product, but does not track the number of observations per bin.



Figure 8 shows the total error (random plus systematic) of the Baker-averaged OCO-2 product versus TCCON for glint and nadir land and ocean comparisons with geometric coincidence. The error is the square root of the standard deviation of (OCO-2 minus TCCON) added in quadrature with the station bias variability. The validation error terms (the co-location error, co-location bias variability, TCCON error, and TCCON sensitivity error) are subtracted in quadrature. The sparse observations over land have errors (~2.5 ppm) much higher than predicted from Eq. 3 (red dashed line), but sparse ocean observations do not have larger errors than denser ocean observations.

The total error is much larger than expected for bins with small $n$ for land, i.e. in locations with low throughput. The error of the $n=1$ bin is 2.5 ppm, whereas the predicted error, using Eq. 3 and Table 2 (using the random error of 1.0 ppm and correlated error of 0.6 ppm from Table 2), is 1.2 ppm. The error drops substantially from $n=1$ through $n=3$, then the error drops more gradually. Even though the errors are higher for low throughput regions, it is possible that value is added because they cover times or locations that would otherwise be missing. However, the predicted errors for the OCO-2 products should be updated according to the above findings, briefly the actual error is shown in Table 6 for land. The ocean errors follow Eq. 3, with random error of 0.6 ppm and systematic error of 0.6 ppm.

Additional types of averaging were tried, e.g. averaging weighted by the predicted error, averaging weighted by the local variability (by setting the error to the standard deviation of each point ± 5 points). Straight averaging versus averaging weighted by the predicted error had a negligible difference (not affecting values rounded to 0.1 ppm) in the random, correlated error and biases.

## 5 Characterizing regional and temporal biases

Regional and seasonal biases are characterized by the mean bias of the observation versus validation data by season and the correlation of error versus distance or time. In Section 5.3.2, the OCO-2 bias term, a stand-in for OCO-2 regional bias, is assimilated to see what effect regional biases have on flux estimates.

### 5.1 Seasonal biases

For seasonal biases, we look at GOSAT, OCO-2, and CarbonTracker versus TCCON or ESRL/DOE aircraft (mainly in the United States) for 3 month groupings (DJF, MAM, JJA, SON) for land or ocean observations, with the mean annual bias subtracted. The TCCON comparisons are grouped by latitude and ocean/land. For the ESRL comparisons, the stations are grouped into geographic clumps of western North America (esp, thd), inland North America (etl, lef, car, hil, sgp), and eastern coastal North America (nha, cma, sca, tgc). The seasonal biases are all less than 0.4 ppm. Significance was tested using the t test. However, the mean biases (even those significant biases) are mostly within the TCCON estimated systematic error (0.4 ppm) and smaller than the OCO-2 or ACOS-GOSAT systematic error estimates of ~0.6 ppm.





There are two locations with both TCCON and aircraft, LEF (Park Falls) and SGP (Lamont) (ETL also has both but not long enough a record so far). The monthly difference of (OCO-2 – TCCON) and (OCO-2- aircraft) was correlated at both Park Falls and Lamont. The correlation at Park Falls was -0.05, and was 0.37 at Lamont.

**5.2 Correlation scales of biases**

This section attempts to quantify the length scales, time scales, and sizes of correlated bias errors. The question we try to answer is: if there is a positive bias in OCO-2 at (latitude1, longitude1, time1), what is the probability that there is a similar positive bias in OCO-2 at (latitude2, longitude2, time2)? Nguyen et al. (2014) looked for the scale length of variability in ACOS-GOSAT data, finding scales of ~15 degrees latitude, ~20 degrees longitude, ~3 days. In this paper we are not looking for the scale length of the signal, but the scale length of the systematic error (OCO-2 minus validation data).

This question is addressed with 3 different methods:

- Correlation of signal versus latitude in remote ocean measurements (where the OCO-2 value is expected to be constant)
- Correlations of biases of (OCO-2 minus ATom) versus latitude
- Correlations of biases of (OCO-2 minus TCCON) versus time

A variogram plots the square of the $XCO_2$ error versus time (days) or distance (latitude) lag. We fit this to an exponential:

$$variogram(lag) = a - be^{-lag/scale} \qquad (4)$$

The resulting values of interest are:

- Random error = sqrt(a-b)
- Correlated error = sqrt(b)
- Scale length = scale, in either distance(latitude) or time (days)

To estimate the size of the correlated region, we use the full-width-half-max for Eq. 4, which is lag = 1.4*scale. Section 5.3 estimates flux errors resulting from regional or temporal biases.

**5.2.1 Correlation of errors versus latitude in remote ocean measurements**

For remote ocean, observations between 90W and 165E longitude and 20S and 10N latitude, in the tropical Pacific, are used.
The true $XCO_2$ on any given day is assumed to be constant within this box, and any variations in the observed data are assumed to be error. The entire box may also have a bias, but this analysis will not quantify the mean bias for group of observations (there is no way to estimate this without using validation data). The data is pre-averaged to 0.1 degree bins to reduce random error.

A variogram showing the co-variance of the corrected $XCO_2$ (binned by 0.1 degrees to reduce error) versus latitude is shown
in Fig. 9. It is useful to keep in mind that values in variograms are co-variances; and a co-variance of 0.2 ppm corresponds to an error of 0.4 ppm.





The fits using Eq. 4 find a random error of 0.1 ppm (which is small because this data was pre-averaged to 0.1 degree bins), and a correlated error of 0.4 ppm. This correlated error should be:

- Smaller than the correlated errors of ocean glint versus TCCON, of 0.6 ppm (from Table 2) because correlated errors contain co-location error and validation error
- Smaller than the corrected systematic error, because the corrected systematic error contains bias error
- Similar to the corrected systematic error (0.5) with the station bias error (0.4) subtracted in quadrature, which is 0.3.

Figure 9 also shows the variogram for the OCO-2 bias correction term (multiplied by 1.6 so that the co-variance at large lags is similar). The bias correction term has a scale length of 0.4 degrees, whereas OCO-2 $XCO_2$ has a scale length of 0.2 degrees. The retrieval parameters that have scale length of 0.5 degrees or less (similar to the scale length of OCO-2) are: albedo_o2a, aod_total, aod_water, dP, co2_grad_del, h2o_scale. The cross-correlation between the above retrieval parameters and $XCO_2$ variations was 0, 0, -0.2, 0.2, 0.4, and 0, respectively. The parameters used in the $XCO_2$ ocean bias correction are dP and co2_grad_del. This analysis shows that aod_water could be considered for ocean bias correction as it has a similar scale length, and the variations in aod_water are anti-correlated to variations in $XCO_2$ for remote ocean locations. The right panel of Fig. 9 shows correlations of variations of $XCO_2$ (corrected), $XCO_2$ (original), the $XCO_2$ bias correction term, and cross-correlations between $XCO_2$ (original and corrected) and the bias correction term. The right panel of Fig. 9 shows that the original $XCO_2$ is correlated with the bias correction term, but the bias corrected $XCO_2$ is *not* correlated with the bias correction term. Figure 9 shows that a) the OCO-2 bias correction term has a similar correlation scale lengths to $XCO_2$, but is not correlated to bias corrected $XCO_2$. The size of the biases (see discussion following Eq. 4) is 1.4*0.2, or ~0.3 degrees.

### 5.2.2 Correlation of errors versus latitude using ATom validation

The same analysis from Section 5.2.1 is done for errors of OCO-2 minus ATom. The idea is the following: if an OCO-2 point has a low bias versus ATom, does a neighboring point also tend to be lower? What is the correlation length of the biases? This is done by binning 0.1 degree bins (to reduce random error) and then calculating the correlation of (OCO-2 – ATom) versus latitude difference. There are 3 ATom results versus latitude with reasonable amounts of data (using geometric criteria), 1N, 1S, 2S. (Dynamic coincidence criteria, which uses very large spatial domains, is not conducive to evaluating spatial biases). The correlation is found to drop off over 5-10 degrees then drop to ~0. Because there are only 3 sets to look at, the accuracy cannot be nailed down more than 5-10 degrees.

### 5.2.3 Correlation of errors versus time using TCCON validation

Time correlation of the error (OCO-2 minus TCCON) is used to calculate correlation of error versus time. The correlation of the bias correction term and the co-location error are also calculated. For geometric coincidences, good fits of the error correlation versus time were not obtained because of the temporal sparseness of the OCO-2 fits versus TCCON. Figure 10 shows the error correlation for land and ocean glint. For land observations co-located using dynamic coincidence criteria versus TCCON, the mean correlation scale (evaluated at every station, then averaged over all stations) was 43 days for





the error and 34 days for the bias term, and the size of the correlated error was 0.8 ppm for the difference and 0.7 ppm for the

bias term. The bias correlation is similar to the error correlation for land but the bias correlation for ocean shows a strong annual cycle, so that the bias correction repeats annually, particularly at stations Burgos, Orleans, Reunion, Tsukuba, and Wollongong. The timescale for the bias correction term is about half for the bias correction term versus the $XCO_2$ error, with the correlated error about the same (0.7 ppm or 0.8 ppm). The timescale of correlations (see discussion around Eq. 4) is 1.4*43 or about 60 days.

Figure 3b, lower panel, shows an example of a temporally persistent low bias in OCO-2 minus TCCON during the period of summer, 2016 lasting several months. CarbonTracker@OCO-2 minus CarbonTracker@TCCON (purple dashed line) suggests that this bias could result from co-location error, because the co-location error also shows a persistent bias, even though the magnitude is smaller. Therefore, the variogram analysis was applied to co-location error (CarbonTracker@OCO-2 minus CarbonTracker@TCCON), finding a correlated error of 0.4 ppm and a timescale of 9 days, suggesting that co-location error

is not responsible for the 43-day correlated error in OCO-2 versus TCCON.

### 5.3 Effect of regional/temporal biases on flux estimates

This section estimates the effects of regional observation biases on flux biases using the relationship that a flux of 2.1 PgC corresponds to a global atmospheric increase of 1 ppm (e.g. Baker et al., 2006). This section calculates how much "fake flux" is caused by biases.

### 5.3.1 Simplistic calculations

We estimate the effect of a bias of 0.5 ppm over a 30 degrees latitude by 60 degrees longitude area using the relationship that 2.1 PgC corresponds to a global atmospheric increase of 1 ppm. This area is about 3% of the surface area of the earth, so a bias of 0.5 ppm only over 3% of the globe would correspond to a flux bias of (0.5 ppm * 2.1 PgC / 1 ppm * 0.03) = 0.03 PgC. The mean absolute value of the fluxes of the TransCom regions from Baker et al. (2006) is 0.5 PgC, so the bias represents a

6% error for an average TransCom region. This gives an idea of the net effect of a 0.5 ppm bias spanning 30 degrees latitude and 60 degrees longitude.

We estimate the effect of a bias over the entire northern hemisphere during April and May. This can be estimated by multiplying the fraction of the yearly airmass affected multiplied by the relationship that 2.1 PgC corresponds to a global atmospheric increase of 1 ppm. The flux bias equals (1/2 the world) * (2/12 months)* 0.5 ppm * (2.1 PgC / 1 ppm) = 87 TgC.

This is a 3% bias of the global flux of 3 PgC. If the flux bias goes into one TransCom region, with a mean flux of 0.5 PgC, it corresponds to an error of 17%.

However, the above describes an unrealistic scenario, where the bias aliases into a single TransCom region. The more likely scenario is, in order to fit a bias that is not consistent with physical fluxes, a pattern of positive and negative fluxes is used to fit the unphysical bias, resulting in positive and negative errors for different TransCom regions. For example, Stephens et al.

(2007) found that model transport error resulted in a positive and negative flux dipole pattern.





Previous work (Kulawik et al., 2013) assimilated a hemispheric summer seasonal bias of 0.3 – 0.5 ppm, finding that this bias caused a pattern of positive and negative flux biases, with positive biases in North America and Asian mid-latitude regions and negative biases in South America, with the sizes of the biases of comparable sizes to the flux updates from GOSAT data itself.

**5.3.2 Assimilation of the OCO-2 bias correction term to estimate systematic flux errors**

This section describes assimilation of the v7 OCO-2 bias correction term. The v7 OCO-2 bias correction term, taken over the v7 record, has size $0.71 \pm 1.10$ ppm for land and $0.03 \pm 0.50$ ppm for ocean. The bias correction consists of a constant bias correction (for land/ocean) plus bias correction that varies regionally and seasonally. A lot of the variable bias correction averages down over small areas, so that averaging over 20 observations, results in a bias of $0.70 \pm 0.68$ ppm for land and 0.04

$\pm 0.35$ ppm for ocean. Section 5.2 finds that the OCO-2 bias correction term has similar correlation length scales as OCO-2 biases, so assimilation of the OCO-2 bias correction term could be used as a crude estimate of the pattern of flux biases. It will *not* lead to conclusions like, "OCO-2 error causes some specific bias in some specific region", but *will* lead to conclusions like, "Regional biases cause errors for Transcom regions on the order of 0.3 GtC/year".

The annual fluxes for every TransCom region are shown in Table 8 for both land nadir (LN) and ocean glint (OG). There is

no overall bias resulting from the bias assimilation, and standard deviation of the flux estimates gives an estimate of the size of the error. Since we estimate that the $XCO_2$ bias from OCO-2 is of similar size to the bias correction term, this can be used to estimate the flux errors resulting from $XCO_2$ regional or seasonal biases. OCO-2 land nadir regional biases cause flux errors on the order of 0.4 PgC/year for land and 0.3 PgC/year for ocean TransCom regions. For land nadir, these biases are fairly persistent from year to year, so comparing fluxes from one year to another results in errors of 0.2 PgC/year for land and 0.1

PgC/year for ocean. These errors are comparable to model transport errors, which are estimated in Basu et al. (2018), Fig. 4. These errors are also comparable to the mean flux magnitude of TransCom regions of 0.2 PgC/year from land and 0.5 PgC/year for ocean.

**6 Discussion and Conclusions**

The systematic errors are estimated for OCO-2 v8 and GOSAT v7.3, which include locally and seasonally correlated error and

regional biases. Although GOSAT has higher random errors than OCO-2 and land has higher random errors than ocean, all modes and both satellites have similar systematic errors, of ~0.6 ± 0.1 ppm. Validation with geometric coincidences vs. dynamic coincidence or validation vs. ESRL aircraft, ATom aircraft, and TCCON results in similar error estimates (after co-location and validation errors are subtracted). From studying averaged quantities, OCO-2 observations from areas with low throughput over land (from pre-processor selection and quality flags) have about twice the error (~2.6 ppm) as areas of high

throughput. The seasonal biases, aggregated over many TCCON stations within latitude bands were usually on the order of 0.2 ppm, indicating no large hemispheric seasonal biases. However, there is an overall low bias in the OCO-2 ocean





observations of -0.7 to -1.0 ppm from ~19N to ~33N. The single observation error, combining the systematic (0.6 ppm) and random errors (0.6 ppm for ocean, 1.0 ppm for land) is 0.8 ppm for ocean and 1.0 ppm for land.

The averaged OCO-2 products have, on average, larger error than is predicted by the product, about 1.2 ppm for land observations and 0.8 ppm for ocean observations. This is partially due to the finding that low throughput areas over land have much higher error, and the purposeful de-emphasis of the high-throughput data. For example, 4% of the Baker-averaged product contain 1 observation, whereas this represents 0.05% of the original data. If averaged products containing at least 20 observations are used, the error estimates are similar to what is expected, based on the number of observations averaged.

The lower partial column from ACOS-GOSAT v7 and OCO-2 v8 was validated, finding systematic errors of 1.1 ppm for both land and ocean. This agrees well with the systematic error estimate from Kulawik et al. (2017) for ACOS-GOSAT v3.5 LMT of 1 ppm. For investigating local effects (e.g. plumes), the random error comes into play, which is 2.6 and 1.0 ppm for land and ocean, respectively, for OCO-2 v8. The higher error for LMT is offset by the larger true variability of the lower partial column, which has 2-3 times the signal variability of $XCO_2$. For signals completely in the lower partial column, the LMT will have 4 times the signal, since it contains 1/4 the airmass of the full column. The LMT product is most useful when the 2 partial columns behave differently.

The raw (un-bias-corrected) products for OCO-2 v8 and ACOS-GOSAT v7.3 were also validated, with the finding that the bias correction improves the overall bias, the systematic error (by about 0.2 ppm for ocean and about 0.5 ppm for land), and the random error (by about 0.5 ppm for land and GOSAT ocean and by 0.2 ppm for OCO-2 ocean).

Since regional and seasonal biases have a detrimental effect on flux estimates, we characterized the time and length scale of biases in corrected $XCO_2$ for OCO-2, finding a small length scale bias (on the order of 0.3 degrees) similarly to Worden et al. (2017), a medium length scale bias (on the order of 5-10 degrees), and a time scale in the bias of ~60 days. The bias correction term, though not correlated with the corrected $XCO_2$ bias, had somewhat similar distance and time scales and sizes. So, to estimate the effect of regional and seasonal biases on flux estimates, the bias correction term from OCO-2 was assimilated and the flux estimates were aggregated on Transcom-3 regions. The assimilation caused flux variations, i.e. positive and negative fluxes which had a standard deviation of 0.4 PgC/year for land regions and 0.3 PgC/yr for ocean regions. The bias correction is somewhat persistent for year-to-year, so that the year-to-year differences in the biases were less than half the error for a particular year.

### Appendix A. Bias correction for LMT

The LMT product and quality flags must be bias corrected for each version, similarly to the $XCO_2$ product. The data was compared to ESRL and DOE aircraft observations co-located using geometric criteria. For ACOS-GOSAT V7.3, the quality flags (in addition to using good quality $XCO_2$ observations) are shown in Table A.1 and A.2, for land and ocean, respectively. The bias term for ACOS-GOSAT v7.3 is:





bias_land = -4.058460 + 0.299454*co2_grad_del -0.544289*dp -1.639720*dofs + 0.030454*glint_angle - 30.231600*aod_ice -4.345870*errorobservation_xco2 -27.450700*aod_dust +6.480650*aod_total -0.315448*aod_oc -

26.739900*aod_water + 1.575730*error_lmt -0.045352* TCWV + 7957.230000* ALBEDO_SLOPE_3 + 0.119988* DP_ABP + 2.306030* H2O_SCALE

bias_ocean = 13.1117000 + CO2_GRAD_DEL*0.5328290 + AOD_OC*10.2381000 + DP_ABP*0.0439909 + ALBEDO_2*-459.8480000 - 3.6511400* AOD_DUST -0.0420218* TCWV + 10.0339000* AOD_SULFATE + 0.3429140* LOGDWS

The bias is subtracted from the raw value to calculate the corrected value.

For OCO-2 v8, the quality flags (in addition to using good quality $XCO_2$ observations) are shown in Table A.3 and A.4, for land and ocean, respectively. The bias term for OCO-2 v8 is:

bias_land = -1.57429000 + 0.25243500* CO2_GRAD_DEL - 0.51346000 * dP + 0.13949800 * S32 - 0.70878200*error_lmt - 9.10000000 * DWS - 0.74062800 * DELTAT + 0.00212755 * SNR_O2A + 18.24700000 *

AOD_SULFATE

bias_ocean = -2.740660 + 0.313480 * co2_grad_del - 0.561932 * dP

The bias is subtracted from the raw value to calculate the corrected value.

## Appendix B.  Station-by-station results

Table B.1 and B.2 show results versus TCCON, ESRL/DOE stations and ATom campaigns for geometric coincidences,

showing the bias, random, and correlated errors shown in Table 2, except for each station used for land (B.1) and ocean (B.2). The standard deviation of these biases is what is called the "Station bias variability" in, e.g. Table 2.  There is no obvious pattern to the land biases, but there is a consistent ocean bias near Saga, Izana, and Burgos between 19 and 33N, of -1.0, -0.9, and -0.7 ppm, respectively (OCO-2 low).  Interestingly, the OCO-2 land data versus Burgos do not have a negative bias.  The other stations that are used for both land and ocean comparisons:  Park Falls, Darwin, Wollongong, and Lauder have biases

less than 0.3 ppm.

Table B.3 shows the overall bias versus TCCON stations for OCO-2 land geometric coincidence for 3 different years (2015, 2016, and 2017).  The question we are interested is whether these biases are persistent year-to-year.  The correlation between 2015 and 2016 biases is -0.27 (0.48 if Eureka and Sodankyla are not included).  The correlation between 2016 and 2017 is 0.54 (0.84 if Sodankyla is not included).

**Author contributions**

Susan Kulawik was responsible for the study design, data analysis, and manuscript writing; Sean Crowell assimilated and analyzed OCO-2 biases; David Baker, Junjie Liu created and advised on averaged products; Kathryn McKain, Colm Sweeney,



Steve Wofsy, Sebastien C. Biraud provided and advised on ATom, ESRL/ATom, ATom, and DOE aircraft observations; Christopher W. O'Dell created and advised on the OCO-2 and ACOS-GOSAT lite products, bias correction, and quality flags;

Paul Wennberg, Debra Wunch, Coleen Roehl, Nicholas Deutscher, Matthäus Kiel, David Griffith, Voltaire Velazco, Justus Notholt, Thorsten Warneke, Christof Petri, Martine De Martine De Mazière, Mahesh Kumar Sha, Ralf Sussmann, Markus Rettinger, Dave Pollard, Isamu Morino, Osamu Uchino, Frank Hase, Dietrich Feist, Sébastien Roche, Kimberly Strong, Rigel Kivi, Laura Iraci, Kawakami Shuji, Manvendra K. Dubey, Eliezer Sepulveda, Omaira Elena Garcia Rodriguez, Yao Te, Pascal Jeseck, Pauli Heikkinen provided and advised on TCCON data; Edward J. Dlugokencky provided and advised on surface

observations; Michael R. Gunson, Annmarie Eldering, David Crisp, Brendan Fisher, Greg Osterman created and advised on OCO-2 and ACOS-GOSAT observations.

**Competing interests**

The authors declare that they have no conflict of interest.

**Data availability**

Data were downloaded from co2.jpl.nasa.gov.

**Acknowledgements**

This research was funded by NASA and performed under BAER Institute's 1029 ARC-CREST cooperative agreement, by NASA Roses ESDR-ERR 10/10-ESDRERR10-0031, "Estimation of biases and errors of CO2 satellite observations from AIRS, GOSAT, SCIAMACHY, TES, and OCO-2" and direct NASA support. TCCON data were obtained from the TCCON

Data Archive, hosted by CaltechDATA: https://tccondata.org/. TCCON at Lamont and Park Falls are funded by NASA grants NNX14AI60G, NNX11AG01G, NAG5-12247, NNG05-GD07G, and the NASA Orbiting Carbon Observatory Program, with the DOE ARM program providing technical support in Lamont and Jeff Ayers providing technical support in Park Falls. The TCCON data at Reunion Island have been acquired by the Royal Belgian Institute for Space Aeronomy with the support of the Université de La Réunion (OPAR team). Financial support was provided through the Belgian 'Science for Sustainable

Development' programme and the ministerial decree FR/35/IC1 to FR/35/IC4. The Darwin and Wollongong TCCON sites are funded by NASA grants NAG512247 and NNG05GD07G and by Australian Research Council grants DP140101552, DP110103118, DP0879468, LE0668470, and LP0562346, and Nicholas Deutscher is supported by and ARC Future Fellowship FT180100327. The Ascension Island TCCON station has been supported by the European Space Agency (ESA) under grant 3-14737 and by the German Bundesministerium für Wirtschaft und Energie (BMWi) under grant 50EE1711C.

The ETL TCCON station is supported by CFI, ORF, NSERC, CSA, and ECCC3. Flights over the Southern Great Plains were



supported by the Office of Biological and Environmental Research of the US Department of Energy under contract no. DE-AC02-05CH11231 as part of the Atmospheric Radiation Measurement (ARM) Program, ARM Aerial Facility (AAF), and Terrestrial Ecosystem Science (TES) Program. SC is supported by NASA Grant NNX15AJ37G. Part of this work was conducted at the Jet Propulsion Laboratory, California Institute of Technology, under contract with the National Aeronautics and Space Administration (NASA).

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



Table 1 Quantification of errors from imperfect validation.  These errors represent systematic, not random, errors.

| | | Land nadir/glint (ppm) | Ocean glint (ppm) |
|---|---|---|---|
| Co-location error | Geometric (CT2017) | 0.4 (0.3, 0.2)* | 0.4 (0.3, 0.2)* |
| | Geometric (CAMS) | 0.2 (0.2, 0.1) | 0.3 (0.3, 0.1) |
| | Geometric (GMAO-50 km) | 0.3 (0.3,0.1)* | 0.5 (0.4, 0.3)* |
| | Geometric (GMAO-7 km) | 0.5 (0.5,0.1)* | 0.5 (0.5, 0.3)* |
| | Dynamic (CT2017) | 0.7 (0.7, 0.2)* | 0.7 (0.7, 0.2)* |
| | Small dynamic (CT2017) | 0.6 (0.6, 0.2)* | 0.6 (0.6, 0.2)* |
| TCCON error (Section 2.1.3) | | 0.4 | 0.4 |
| Averaging Kernel error (TCCON) | | 0.1 | 0.1 |
| Aircraft error (Section 2.1.2) | | 0.1 ($XCO_2$) 0.4 (LMT) | 0.1 ($XCO_2$) 0.4 (LMT) |
| Aircraft profile extension to TOA (ATom) | | 0.1 | 0.1 |
| Aircraft profile extension to TOA (ESRL/DOE) | | 0.1 | 0.1 |
| Aircraft profile sampling and extension to surface (ESRL/DOE) | | 0.1 | 0.1 |

* The co-location error is shown as the total systematic error with the correlated and station bias errors in parentheses)






Table 2 random and correlated errors and bias variability for geometric coincidence criteria and non-sequential averaging. Numbers in parentheses are the co-location error estimates. The Systematic error adds (in quadrature) the correlated error and the station bias. The corrected systematic error removes co-location and validation errors. Numbers are reported, rounded to the nearest 0.1 ppm. The non-bold entries are intermediate values.

| | Mean bias (ppm) | Random error (ppm) | Correlated error (ppm) | Station bias error (ppm) | Systematic error (ppm) | Co-location corrected systematic error (ppm) | Corrected systematic error (ppm)** |
|---|---|---|---|---|---|---|---|
| **ACOS-GOSAT (dynamic)** | | | | | | | |
| Land (TCCON) | **-0.1 (-0.1)** | **1.7 (0.6)** | 0.8 (0.6) | 0.7 (0.1) | 1.1 (0.6) | 0.9 | **0.7** |
| Land (ESRL/DOE aircraft) | **-0.1 (-0.1)** | **1.6 (0.7)** | 0.8 (0.7) | 0.5 (0.2) | 0.9 (0.7) | 0.6 | **0.5** |
| Ocean (TCCON) | **-0.3 (0.0)** | **1.1 (0.3)** | 0.9 (0.5) | 0.4 (0.1) | 1.0 (0.5) | 0.8 | **0.6** |
| **OCO-2 (geometric coincidence)** | | | | | | | |
| Land nadir (TCCON) | **0.0 (0.0)** | **1.0 (0.3)** | 0.8 (0.3) | 0.6 (0.1) | 1.0 (0.3) | 0.9 | **0.8** |
| land glint (TCCON) | **-0.1 (-0.1)** | **1.1 (0.3)** | 0.7 (0.4) | 0.6 (0.3) | 0.9 (0.5) | 0.8 | **0.6** |
| All land (TCCON) | **-0.1 (-0.1)** | **1.1 (0.3)** | 0.7 (0.3) | 0.6 (0.2) | 0.9 (0.4) | 0.8 | **0.6** |
| All land (TCCON) (20-ave) | **-0.1(-0.1)** | **0.7 (0.3)** | 0.6 (0.2) | 0.6 (0.2) | 0.9 (0.4) | 0.8 | **0.6** |
| Ocean glint (TCCON) | **-0.2 (0.1)** | **0.5 (0.2)** | 0.6 (0.4) | 0.4 (0.2) | 0.7 (0.4) | 0.6 | **0.3** |
| Ocean glint (TCCON) (20-ave) | **-0.2 (0.1)** | **0.6 (0.2)** | 0.7 (0.4) | 0.4 (0.2) | 0.8 (0.4) | 0.7 | **0.4** |
| Ocean glint (ATom) | **-0.7 (-0.1)** | **0.8 (0.3)** | 0.1 (0.2) | 0.5 (0.3) | 0.5 (0.4) | 0.4 | **0.3** |
| **OCO-2 (dynamic coincidence)** | | | | | | | |
| Land nadir (TCCON)* | **-0.2 (-0.1)** | **0.9 (0.6)** | 0.9 (0.7) | 0.5 (0.2) | 1.0 (0.7) | 0.7 | **0.5** |
| land glint (TCCON)* | **-0.2 (-0.2)** | **1.0 (0.7)** | 0.9 (0.7) | 0.6 (0.3) | 1.1 (0.8) | 0.8 | **0.6** |
| All land (TCCON)* | **-0.2 (-0.1)** | **1.0 (0.6)** | 0.9 (0.7) | 0.5 (0.2) | 1.0 (0.7) | 0.7 | **0.5** |





| | | | | | | | |
|---|---|---|---|---|---|---|---|
| All land (ESRL/DOE aircraft) | **0.0 (0.0)** | **1.2 (0.6)** | 0.9 (0.8) | 0.5 (0.3) | 1.0 (0.9) | 0.6 | **0.5** |
| Ocean glint (TCCON) | **-0.2 (0.1)** | **0.6 (0.5)** | 0.9 (0.7) | 0.6 (0.2) | 1.1 (0.7) | 0.8 | **0.6** |
| Ocean glint (ESRL/DOE aircraft ) | -0.1 (0.2) | **0.7 (0.5)** | 1.3 (0.8) | 0.3 (0.2) | 1.3 (0.8) | 1.0 | **1.0[+]** |
| **CT2016/CT2017NRT (geometric)** | | | | | | | |
| Land nadir (TCCON) | **-0.2** | **0.3** | 0.3 | 0.5 | 0.6 | 0.6 | **0.4** |
| Land glint (TCCON) | **-0.1** | **0.3** | 0.4 | 0.5 | 0.6 | 0.6 | **0.5** |
| Ocean (TCCON) | **-0.2** | **0.2** | 0.4 | 0.4 | 0.6 | 0.6 | **0.4** |
| Ocean (ATom) | **-0.4** | **0.3** | 0.2 | 0.6 | 0.6 | 0.6 | **0.6** |


* The TCCON dynamic results are done with 20-observation averages due to huge number of co-locations for dynamic coincidence, resulting in smaller random error (offset by higher random co-location error)

** The far right column is the 2nd from the right column with the validation errors from Table 1 subtracted in quadrature or directly subtracted (for the averaging kernel sensitivity error).

[+]Likely high due to comparisons of ocean versus continental observations.



Table 3. OCO-2 and GOSAT errors without bias correction. These can be compared to the same cases from Table 2. The non-bold entries are intermediate values.

| | Mean bias (ppm) | Random error (ppm) | Correlated error (ppm) | Station bias error (ppm) | Systematic error (ppm) | Co-location corrected systematic error (ppm) | Corrected systematic error (ppm)** |
|---|---|---|---|---|---|---|---|
| **GOSAT (dynamic coincidence)** | | | | | | | |
| land (TCCON) | **0.5 (-0.1)** | **2.0 (0.7)** | 0.9 (0.6) | 1.1 (0.1) | 1.4 (0.6) | 1.3 | **1.1** |
| Ocean (TCCON) | **0.1 (0.0)** | **1.5 (0.3)** | 1.0 (0.5) | 0.4 (0.1) | 1.1 (0.5) | 0.9 | **0.8** |
| **OCO-2 (geometric coincidence)** | | | | | | | |
| land (TCCON) | **-2.1 (-0.1)** | **1.6 (0.3)** | 1.0 (0.3) | 0.8 (0.2) | 1.3 (0.4) | 1.2 | **1.0** |
| Ocean (TCCON) | **-2.5 (0.0)** | **0.8 (0.2)** | 0.8 (0.4) | 0.5 (0.2) | 0.9 (0.4) | 0.8 | **0.7** |

** Systematic error with co-location error and validation error subtracted






Table 4. Validation of lower-tropospheric OCO-2 versus aircraft (listed in Table 2) and remote ocean surface measurements.
The non-bold entries are intermediate values.

| | Mean bias (ppm) | Random error (ppm) | Correlated error (ppm) | Station bias error (ppm) | Systematic error (ppm) | Co-location corrected systematic error (ppm) | Corrected systematic error (ppm)** |
|---|---|---|---|---|---|---|---|
| **GOSAT LMT (dynamic coincidence)** | | | | | | | |
| land (aircraft) | **-0.3 (-0.2)** | **2.8 (1.8)** | 2.2 (2.0) | 0.9 (0.6) | 2.4 (2.1) | 1.1 | **1.1** |
| Ocean (aircraft) | **-0.9 (1.4)** | **2.0 (0.9)** | 2.9 (2.5) | 1.8 (1.7) | 3.4 (3.0) | 1.6 | **1.6** |
| Ocean (aircraft, rta only) | **0.1 (0.0)** | **1.6 (0.3)** | 0.7 (0.3) | | | | **~0.7** |
| Ocean (surface) | **-0.5 (0.3)** | **1.7 (0.6)** | 1.2 (0.7) | 0.8 (0.3) | 1.4 (0.8) | 1.2 | **1.2** |
| **OCO-2 LMT (dynamic coincidence)** | | | | | | | |
| land (ESRL/DOE aircraft) | **0.1 (-0.2)** | **2.6 (1.9)** | 2.3 (2.2) | 0.9 (0.7) | 2.5 (2.3) | 0.9 | **0.8** |
| Ocean (ESRL/DOE aircraft) | **0.0 (0.4)** | **1.6 (1.1)** | 3.4 (2.1) | 1.5 (0.5) | 3.7 (2.2) | 3.0 | **3.0** |
| Ocean (aircraft, rta only) | **0.0 (0.1)** | **1.2 (0.3)** | 1.0 (0.3) | | | | **~1.0** |
| Ocean (ATom aircraft) (dynamic) | **-0.8 (-0.4)** | **2.2 (1.3)** | 0.8 (0.7) | 0.8 (0.8) | 1.1 (1.1) | - | **<1.1** |
| Ocean (surface, geometric coinc.) | **-0.3 (-0.1)** | **1.0 (0.4)** | 1.2 (0.4) | 0.8 (0.3) | 1.4 (0.5) | 1.4 | **1.3** |




Table 5. Standard deviation and bias variability for OCO-2 averaged products using geometric coincidence. The same set of stations is used as in Table 2. The "Baker (# > 20)" entry considers only bins that have at least 20 observations. The Systematic error combines the standard deviation (which assumes is correlated error) and station bias, and subtracts the co-location error and validation error, similarly to what was done in Table 2.


| | Bias (ppm) | Stdev (ppm) | Station bias (ppm) | Systematic error (corrected) (ppm) | Predicted error (ppm) |
|---|---|---|---|---|---|
| Land nadir | | | | | |
| Lite 20-obs average | **0.1 (0.0)** | 1.0 (0.4) | 0.6 (0.1) | **1.0** | 0.7[*] |
| Baker (# > 20) | **0.0 (-0.1)** | 0.9 (0.5) | 0.6 (0.2) | **0.8** | 0.7[**] |
| Baker | **0.4 (0.0)** | 1.3 (0.4) | 0.5 (0.1) | **1.4** | 0.7[**] |
| Liu | **0.4 (0.0)** | 1.4 (0.6) | 0.5 (0.1) | **1.3** | 1.1[***] |
| Land glint | | | | | |
| Lite 20-obs average | **0.0 (-0.1)** | 1.0 (0.5) | 0.5 (0.3) | **0.8** | 0.6[*] |
| Baker (# > 20) | **0.0 (-0.1)** | 0.9 (0.5) | 0.6 (0.2) | **0.8** | 0.7[**] |
| Baker | **0.3 (-0.1)** | 1.3 (0.5) | 0.6 (0.2) | **1.2** | 0.7[**] |
| Liu | **0.3 (-0.1)** | 1.3 (0.6) | 0.5 (0.2) | **1.2** | 1.1[***] |
| Ocean glint | | | | | |
| Lite 20-obs | **-0.2 (0.0)** | 0.8 (0.5) | 0.4 (0.2) | **0.6** | 0.5[*] |
| Baker (# > 20) | **-0.2 (0.0)** | 0.8 (0.4) | 0.4 (0.2) | **0.6** | 0.5[**] |
| Baker | **-0.2 (0.0)** | 0.9 (0.4) | 0.4 (0.2) | **0.8** | 0.6[**] |
| Liu | **-0.2 (0.0)** | 0.9 (0.4) | 0.4 (0.1) | **0.8** | 0.7[***] |

[*]Using Eq. 3 and values from Table 2 (OCO-2 geometric coincidence results).[**]Field xco2_uncertainty.[***]Field XCO2_STD+UNCERTAINTY





Table 6. The actual errors for the Baker-average OCO-2 products. The errors reflect the fact that areas with sparser good quality observations have higher errors.


| Number | Error (ppm) |
|--------|-------------|
| 1 | 2.5 |
| 2 | 1.7 |
| 3-15 | 1.3 |
| 16-25 | 1.2 |
| 26-42 | 1.1 |
| 43-70 | 1.0 |
| 71-113 | 0.9 |
| 114-189 | 0.8 |
| 190-240 | 0.7 |


Table 7. Mean seasonal biases versus aircraft and TCCON for ocean and land. The stations comprising the averages are: Bialystok, Karlsruhe, Orleans, Garmisch; Park Falls, Lamont, Anmyeondo, Dryden; Darwin, Wollongong, Lauder125 for TCCON land, and Saga, Izana; Ascension, Darwin, Reunion, Wollongong, Lauder for TCCON ocean; dnd, thd for western North America, etl,lef,car,hil,sgp for inland North America, and nha, cma, sca, and tgc for eastern coastal North America. The bold entries indicate statistically significant biases (using the $t$ test) of at least 0.2 ppm magnitude.

| Category | CT | | | | ACOS-GOSAT | | | | OCO-2 | | | |
|---|---|---|---|---|---|---|---|---|---|---|---|---|
| | DJF | MAM | JJA | SON | DJF | MAM | JJA | SON | DJF | MAM | JJA | SON |
| NH > 45N (TCCON land) | -0.1 | 0.3 | -0.1 | -0.1 | -0.1 | -0.1 | 0.0 | **0.2** | 0.1 | 0.0 | -0.1 | 0.0 |
| NH 0-45N (TCCON land) | 0.1 | -0.2 | 0.2 | -0.2 | 0.3 | **-0.3** | 0.1 | -0.1 | **0.4** | -0.1 | -0.2 | -0.1 |
| West coast N. America (ESRL land) | 0.0 | **0.2** | 0.1 | -0.3 | 0.2 | -0.3 | **0.2** | -0.1 | 0.2 | 0.0 | 0.1 | -0.3 |
| Inland N. America (ESRL land) | -0.4 | **0.4** | 0.2 | **-0.2** | 0.0 | -0.1 | 0.1 | -0.1 | -0.2 | **0.3** | 0.0 | -0.1 |
| East coast N. America (ESRL land) | 0.1 | 0.1 | -0.2 | 0.0 | **0.2** | -0.1 | -0.1 | -0.1 | 0.2 | **0.2** | -0.4 | 0.0 |
| NH 0-45N (TCCON ocean) | -0.2 | 0.0 | 0.3 | -0.0 | -0.4 | 0.3 | 0.2 | 0.0 | -0.1 | 0.1 | -0.1 | 0.1 |
| SH (TCCON land) | -0.3 | **0.2** | 0.2 | -0.1 | -0.2 | 0.0 | **0.3** | -0.1 | -0.2 | 0.1 | -0.1 | 0.2 |
| SH (TCCON ocean) | **-0.3** | 0.3 | 0.3 | -0.2 | **-0.2** | -0.2 | 0.4 | 0.1 | **-0.3** | -0.1 | 0.3 | 0.1 |





Table 8.  Assimilation of OCO-2 XCO$_2$ bias correction term for land nadir (LN) and ocean glint (OG) annual fluxes.

| # | TransCom region | LN Annual flux 2015 (PgC) | LN Annual flux 2016 (PgC) | LN Annual flux 2016-2015 (PgC) | OG Annual flux 2015 (PgC) | OG Annual flux 2016 (PgC) | OG Annual flux 2016-2015 (PgC) |
|---|---|---|---|---|---|---|---|
| 1 | North American Boreal | -0.06 | -0.18 | -0.12 | 0.03 | -0.07 | -0.1 |
| 2 | North American Temperate | -0.26 | -0.62 | -0.36 | 0.01 | 0.25 | 0.24 |
| 03a | N. Tropical South America | -0.06 | -0.17 | -0.11 | -0.18 | -0.13 | 0.05 |
| 03b | S. Tropical South America | -0.47 | -0.47 | 0.00 | 0.07 | 0.19 | 0.12 |
| 4 | South American Temperate | -0.56 | -0.79 | -0.23 | 0.12 | -0.04 | -0.16 |
| 05a | Temp. N. extratrop. Africa | 0.32 | 0.35 | 0.03 | 0.04 | 0.03 | -0.01 |
| 05b | Northern Tropical Africa | 0.72 | 0.41 | -0.31 | 0.09 | -0.03 | -0.12 |
| 06a | Southern Tropical Africa | -0.33 | -0.28 | 0.05 | -0.14 | 0.07 | 0.21 |
| 06b | Temp. S. extratrop. Africa | 0.12 | 0.19 | 0.07 | 0.15 | 0.05 | -0.1 |
| 7 | Eurasia Boreal | -0.09 | 0.20 | 0.29 | -0.05 | -0.04 | 0.01 |
| 8 | Eurasia Temperate | -0.33 | -0.19 | 0.14 | 0.33 | 0.16 | -0.17 |
| 09a | Northern Tropical Asia | -0.65 | -0.59 | 0.06 | -0.09 | -0.07 | 0.02 |
| 09b | Southern Tropical Asia | -0.16 | -0.14 | 0.02 | -0.27 | -0.3 | -0.03 |
| 10a | Tropical Australia | 0.14 | 0.12 | -0.02 | -0.09 | -0.08 | 0.01 |
| 10b | Temperate Australia | 0.33 | 0.36 | 0.03 | 0.13 | 0.05 | -0.08 |
| 11 | Europe | 0.38 | 0.46 | 0.08 | 0.15 | 0.03 | -0.12 |
| | **Land mean** | **-0.06** | **-0.08** | **-0.02** | **0.02** | **0.00** | **-0.01** |
| | **Land stdev** | **0.38** | **0.40** | **0.17** | **0.15** | **0.13** | **0.12** |
| 12 | North Pacific Temperate | -0.38 | -0.58 | -0.2 | 0.07 | 0.24 | 0.17 |
| 13 | West Pacific Tropical | -0.08 | -0.07 | 0.01 | -0.17 | -0.14 | 0.03 |
| 14 | East Pacific Tropical | -0.08 | -0.11 | -0.03 | -0.28 | -0.11 | 0.17 |
| 15 | South Pacific Temperate | 0.03 | 0.00 | -0.03 | -0.04 | -0.05 | -0.01 |
| 16 | Northern Ocean | 0.09 | 0.11 | 0.02 | 0.04 | -0.02 | -0.06 |



| | | | | | | | |
|---|---|---|---|---|---|---|---|
| 17 | North Atlantic Temperate | 0.74 | 0.66 | -0.08 | 0.08 | 0.08 | 0 |
| 18 | Atlantic Tropical | -0.01 | -0.02 | -0.01 | 0 | 0.1 | 0.1 |
| 19 | South Atlantic Temperate | 0.03 | 0.00 | -0.03 | 0.08 | -0.05 | -0.13 |
| 20 | Southern Ocean | 0.24 | 0.30 | 0.06 | 0.34 | 0.22 | -0.12 |
| 21 | Indian Tropical | 0.10 | 0.14 | 0.04 | -0.1 | -0.09 | 0.01 |
| 22 | South Indian Temperate | 0.24 | 0.23 | -0.01 | 0.04 | -0.02 | -0.06 |
| 23 | Not optimized | 0.02 | 0.03 | 0.01 | 0.03 | 0.02 | -0.01 |
| | **Ocean mean** | **0.08** | **0.06** | **-0.02** | **0.01** | **0.02** | **0.01** |
| | **Ocean stdev** | **0.26** | **0.29** | **0.07** | **0.15** | **0.12** | **0.10** |
| | **All mean** | **0.00** | **-0.02** | **-0.02** | **0.01** | **0.01** | **-0.01** |
| | **All stdev** | **0.34** | **0.36** | **0.13** | **0.15** | **0.13** | **0.11** |



Table A.1 ACOS-GOSAT v7.3 LMT Land quality flags

|  | General | O2A | Weak | Strong |
|---|---|---|---|---|
| Sounding airmass | < 3.2 |  |  |  |
| Co2_grad_del | < 100, > -50 |  |  |  |
| H2o_scale | < 1.3, > 0.7 |  |  |  |
| snr |  | > 80 | > 70 | > 150 |
| Aod_oc | < 0.13 |  |  |  |
| Reduced_chi_squared |  | < 1.3 | < 1.5, > 0.7 | < 1.5 |
| Aod_seasalt | < 0.06 |  |  |  |
| S32 | > 0.22 |  |  |  |
| tcwv_uncertainty | < 0.4 |  |  |  |




Table A.2 ACOS-GOSAT v7.3 LMT Ocean quality flags

|  | General | O2A | Weak | Strong |
|---|---|---|---|---|
| Sounding airmass | < 2.5 |  |  |  |
| H2o_scale | < 1.4 |  |  |  |
| Aod_seasalt | < 0.4 |  |  |  |
| Aod_ice | < 0.1 |  |  |  |
| S31 | > 0.42, < 0.57 |  |  |  |
| S32 | < 0.61 |  |  |  |
| Albedo_slope |  |  | > 0 |  |
| Retrieved surface pressure | < 1025 |  |  |  |
| Predicted error | < 7 ppm |  |  |  |
| Pre-processor H2o_ratio | > 0.92, < 1.02 |  |  |  |





**A.2 OCO-2 v8**

Table A.3 OCO-2 v8 LMT Land quality flags

|  | General | O2A | Weak | Strong |
|---|---|---|---|---|
| deltaT | < 2 |  |  |  |
| Eof3_3_rel | < 1 |  |  |  |
| snr |  | < 600 |  |  |
| Aod_bc | < 0.01 |  |  |  |
| Land_fraction | > 0.99 |  |  |  |



Table A.4 OCO-2 v8 LMT Ocean quality flags

|  | General | O2A | Weak | Strong |
|---|---|---|---|---|
| Albedo |  | < 0.05 |  |  |
| Aod_dust | < 0.04 |  |  |  |
| Aod_seasalt | < 0.08 |  |  |  |
| Aod_surfate | < 0.10 |  |  |  |
| aod_total | < 0.10 |  |  |  |
| aod_oc | < 0.02 |  |  |  |
| aod_water | < 0.02 |  |  |  |


Table B.1. Station-by-station results for OCO-2 land versus TCCON for geometric coincidences. The mean bias, random and correlated errors are shown with the co-location estimates for each shown in ().

| TCCON station | Latitude (deg) | Longitude (deg) | bias(col) (ppm) | random(col) (ppm) | corr(col) (ppm) |
|---|---|---|---|---|---|
| Eureka | 80 | -86 | -1.5(0.1) | 2.1(0.1) | 0.5(0.1) |
| Sodankyla | 67 | 27 | 0.6(0.2) | 1.0(0.6) | 0.8(0.4) |
| East Trout Lake | 54 | -105 | -0.2(-0.2) | 1.1(0.6) | 0.9(0.6) |
| Bialystok | 53 | 23 | 0.3(0.1) | 1.1(0.4) | 0.7(0.3) |
| Karlsruhe | 49 | 8 | 1.0(-0.5) | 1.1(0.3) | 0.8(0.5) |
| Orleans | 48 | 2 | 0.4(0.1) | 1.0(0.3) | 0.6(0.4) |
| Garmisch | 47 | 11 | 0.1(-0.1) | 1.2(0.4) | 1.4(0.6) |
| Park Falls | 46 | -90 | -0.2(0.1) | 1.0(0.6) | 0.6(0.5) |
| Lamont | 37 | -97 | 0.0(0.0) | 0.8(0.3) | 0.6(0.3) |
| Anmyeondo | 37 | 126 | -0.1(-0.3) | 1.3(0.3) | 0.6(0.3) |
| Dryden | 35 | -118 | 0.5(-0.1) | 1.1(0.2) | 1.1(0.3) |
| Hefei | 32 | 117 | -0.8(-0.2) | 0.8(0.4) | 1.6(0.3) |
| Burgos | 19 | 121 | 0.0(0.0) | 1.3(0.1) | 0.1(0.0) |
| Manaus | -3 | -61 | -1.2(-0.1)* | 1.9(0.2) | 0.0(0.5) |
| Darwin | -12 | 131 | -0.1(0.0) | 0.7(0.1) | 0.7(0.1) |
| Wollongong | -34 | 151 | -0.3(0.0) | 0.9(0.1) | 0.6(0.2) |
| Lauder125 | -45 | 170 | 0.0(0.0) | 1.0(0.1) | 0.7(0.1) |
| Aircraft station | Latitude (deg) | Longitude (deg) | bias(col) (ppm) | random(col) (ppm) | corr(col) (ppm) |
| pfa | 65 | -149 | 0.6(0.5) | 1.3(0.8) | 0.9(0.4) |
| etl | 54 | -105 | 0.5(0.5) | 1.2(0.4) | 1.0(1.3) |
| dnd | 47 | -99 | -0.6(0.1) | 1.1(0.7) | 1.1(1.2) |
| nsa | 70 | -153 | 0.7(0.4) | 1.3(0.7) | 0.8(0.8) |
| lef | 46 | -90 | -0.2(-0.4) | 1.1(0.6) | 1.5(0.8) |
| nha | 43 | -70 | 0.1(-0.2) | 1.1(0.7) | 0.7(0.9) |
| thd | 41 | -124 | 0.2(0.4) | 1.5(0.7) | 1.4(1.0) |
| car | 41 | -104 | -0.2(0.0) | 1.3(0.7) | 0.9(0.4) |
| hil | 40 | -88 | -0.6(-0.3) | 1.2(0.7) | 0.8(0.7) |
| cma | 39 | -74 | 0.0(-0.2) | 1.1(0.7) | 0.8(0.8) |
| sgp | 37 | -97 | -0.5(-0.2) | 1.4(0.7) | 0.7(0.7) |
| sca | 33 | -80 | 0.0(-0.2) | 1.2(0.6) | 0.9(0.6) |
| tgc | 28 | -97 | -0.4(-0.3) | 1.2(0.6) | 0.5(0.5) |

* Sparse number of observations





Table B.2. Station-by-station results for OCO-2 ocean versus TCCON. The mean bias, random and correlated errors are shown with the co-location estimates for each shown in ().

| TCCON station | Latitude (deg) | Longitude (deg) | bias(col) (ppm) | random(col) (ppm) | corr(col) (ppm) |
|---|---|---|---|---|---|
| Park Falls | 46 | -90 | -0.1(0.5) | 0.8(0.7) | 0.8(0.8) |
| Rikubetsu | 43 | 144 | 0.1(-0.1) | 0.7(0.0) | 0.6(1.1) |
| Tsukuba125 | 36 | 140 | -0.4(-0.1) | 0.9(0.5) | 0.9(0.5) |
| Saga | 33 | 130 | -1.0(0.1) | 0.8(0.4) | 0.7(0.5) |
| Izana | 28 | -16 | -0.9(0.0) | 0.6(0.1) | 0.5(0.3) |
| Burgos | 19 | 121 | -0.7(0.3) | 0.6(0.1) | 0.7(0.2) |
| Ascension | -8 | -14 | 0.1(0.0) | 0.6(0.1) | 0.6(0.1) |
| Darwin | -12 | 131 | 0.1(0.0) | 0.5(0.1) | 0.7(0.2) |
| Reunion | -21 | 55 | 0.0(0.0) | 0.6(0.1) | 0.8(0.1) |
| Wollongong | -34 | 151 | 0.2(0.0) | 0.5(0.1) | 0.6(0.3) |
| Lauder125 | -45 | 170 | 0.1(0.0) | 0.5(0.1) | 0.4(0.1) |
| ESRL station | Latitude (deg) | Longitude (deg) | bias(col) (ppm) | random(col) (ppm) | corr(col) (ppm) |
| Rarotonga | -21 | -160 | -0.2(0.0) | 0.5(0.1) | 0.4(0.1) |
| ATom campaign | Date start | Date end | bias(col) (ppm) | random(col) (ppm) | corr(col) (ppm) |
| 1S | 8/3/16 | 8/22/26 | -1.2(-0.5) | 1.0(0.5) | 0.1(0.4) |
| 1 | 8/22/16 | 8/22/16 | -0.4(0.3) | 0.7(0.6) | 0.0(0.3) |
| 2S | 2/1/17 | 2/10/17 | -0.7(0.0) | 0.6(0.1) | 0.2(0.1) |
| 2 | 1/26/17 | 1/29/17 | -1.2(0.0) | 0.7(0.1) | 0.0(0.0) |
| 2N | 2/10/17 | 2/15/17 | -0.1(-0.1) | 0.8(0.3) | 0.1(0.1) |



Table B.3 OCO-2 biases versus TCCON for 2015, 2016, and 2017 for OCO-2 geometric land coincidences with co-location
bias in ().

| | Lat | Lon | 2015 bias (col) | 2016 bias (col) | 2017 bias (col) |
|---|---|---|---|---|---|
| Eureka | 80 | -86 | -1.5(0.1) | 0.8(0.2) | |
| Sodankyla | 67 | 27 | 1.0(0.0) | -1.3(0.0) | 0.2(0.3) |
| Bialystok | 53 | 23 | 0.3(0.2) | 0.1(-0.1) | 0.0(0.2) |
| Karlsruhe | 49 | 8 | 1.1(-0.6) | 0.8(-0.4) | 1.3(-0.3) |
| Orleans | 48 | 2 | 0.7(0.0) | 0.5(0.1) | 0.1(-0.1) |
| Garmisch | 47 | 11 | 0.2(-0.2) | 0.0(-0.1) | 0.4(-0.1) |
| Parkfalls | 46 | -90 | -0.1(0.1) | -0.4(0.0) | -0.2(0.2) |
| Lamont | 37 | -97 | 0.2(0.0) | 0.0(0.0) | -0.2(0.0) |
| Anmyeondo | 37 | 126 | -0.1(-0.3) | 0.0(-0.2) | |
| Dryden | 35 | -118 | 0.5(-0.1) | 0.3(-0.1) | |
| Hefei | 32 | 117 | 0.5(-0.3) | -1.1(-0.2) | |
| Darwin | -12 | 131 | 0.3(0.0) | -0.7(0.0) | -0.5(0.0) |
| Wollongong | -34 | 151 | -0.3(-0.1) | -0.3(-0.1) | -0.5(0.0) |
| Lauder125 | -45 | 170 | -0.1(0.0) | -0.2(0.0) | 0.0 (0.0) |





Table B.4 OCO-2 v8 biases versus TCCON for ocean geometric coincidences showing stations with comparisons all 3 years, with co-location bias in ().

| | Lat | Lon | 2015 bias (col) | 2016 bias (col) | 2017 bias (col) |
|---|---|---|---|---|---|
| Parkfalls | 46 | -90 | 0.0(0.6) | -0.3(0.4) | -0.6(0.0) |
| Rikubetsu | 43 | 144 | 0.3(-0.3) | 0.3(-0.4) | -0.2(0.3) |
| Tsukuba125 | 36 | 140 | -0.4(-0.2) | 0.1(0.2) | -0.7(-0.2) |
| Saga | 33 | 130 | -0.8(0.0) | -1.2(-0.1) | -1.1(0.3) |
| Izana | 28 | -16 | -0.6(-0.2) | -0.7(0.1) | -1.1(0.0) |
| Ascension | -8 | -14 | 0.1(0.0) | 0.1(-0.1) | -0.2(0.0) |
| Darwin | -12 | 131 | 0.5(0.1) | -0.4(0.0) | -0.5(0.0) |
| Reunion | -21 | 55 | 0.3(0.0) | 0.2(0.0) | -0.2(0.0) |
| Wollongong | -34 | 151 | 0.7(0.0) | -0.2(0.0) | 0.0(-0.1) |
| Lauder125 | -45 | 170 | 0.2(0.0) | -0.2(0.0) | 0.1(0.0) |






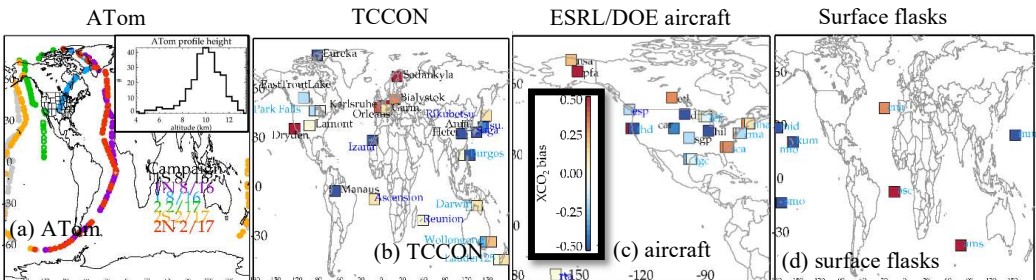

**Figure 1: Observation locations for (a) ATom, (b) TCCON, (c) ESRL and DOE aircraft, and (d) "remote" surface observations. ATom campaigns are designated by color, with the profiles extending above 12 km with a "+" placed, with inset showing the distribution of ATom heights. For (b), (c), and (d) the colors designate the bias of OCO-2 XCO₂ at each site. If a site is used for land and ocean OCO-2 validation, the left side of bias is for land and right side of the bias is for ocean.**






**Figure 2: ACOS-GOSAT co-located to TCCON using geometric and dynamic coincidence criteria. Monthly averages are shown for**
**ACOS-GOSAT (red), TCCON (pink), the ACOS-GOSAT prior (blue/green), and the CarbonTracker model (purple dotted) (n=**
**shows the number of months containing co-locations). The lower plots show differences of ACOS-GOSAT (red) and the prior**





**(blue/green) difference versus TCCON. The dashed purple line on the lower plots shows the difference between the CarbonTracker model co-located with GOSAT observation location/times minus the CarbonTracker model co-located with TCCON observation/times (to quantify the error introduced by co-location).**




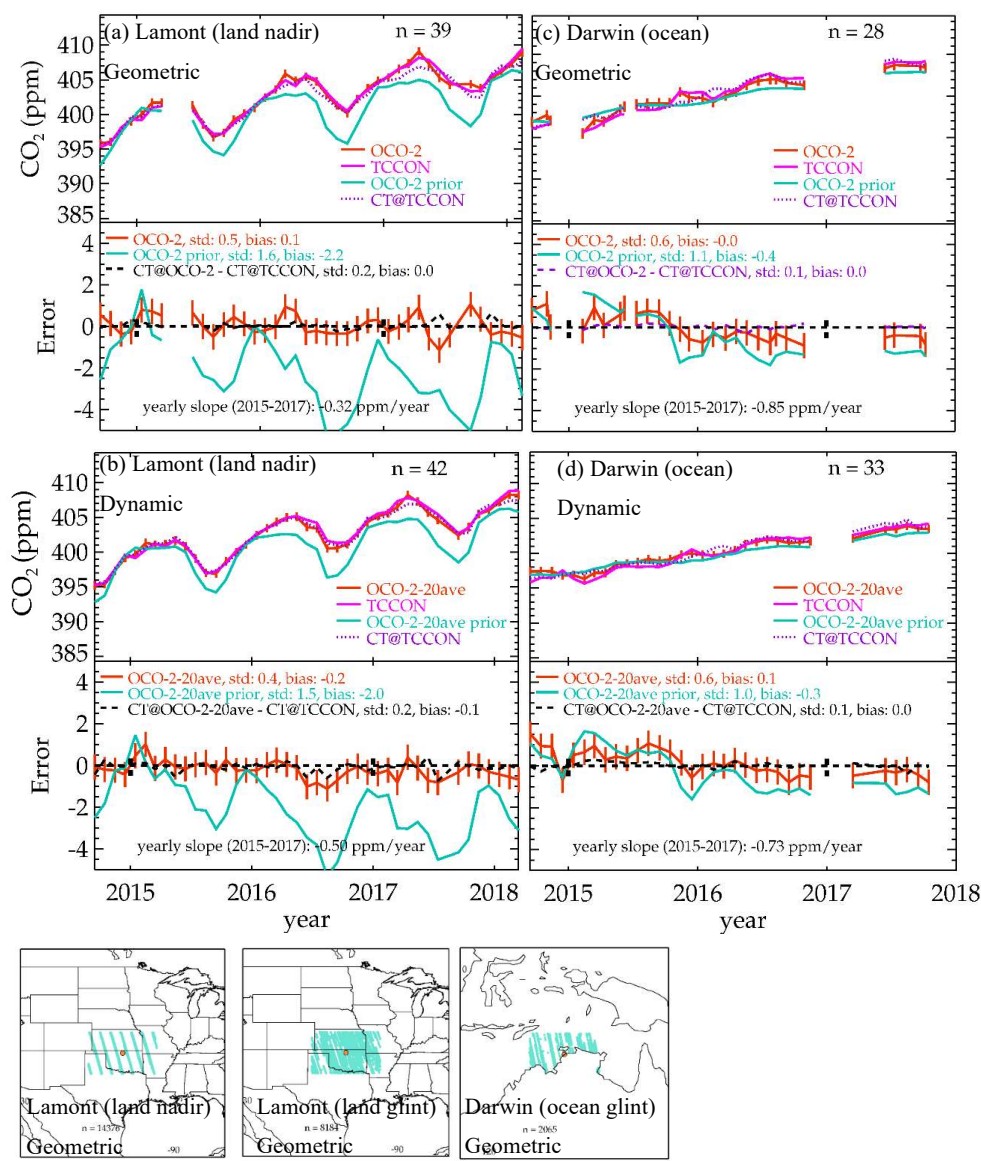





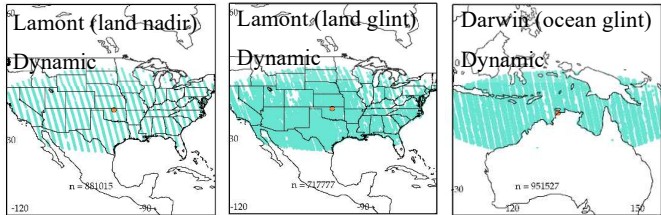

**Figure 3: OCO-2 co-located with TCCON using geometric and dynamic coincidence criteria. Monthly averages are shown for OCO-2 (red), TCCON (pink), the OCO-2 prior (blue/green), and the CarbonTracker model (purple dotted). The lower plots show differences of OCO-2 (red) and the prior (blue/green) difference versus TCCON. The dashed purple line on the lower plots shows the difference between the CarbonTracker model co-located to OCO-2 observation location/times minus the CarbonTracker model co-located to TCCON observation/times.**



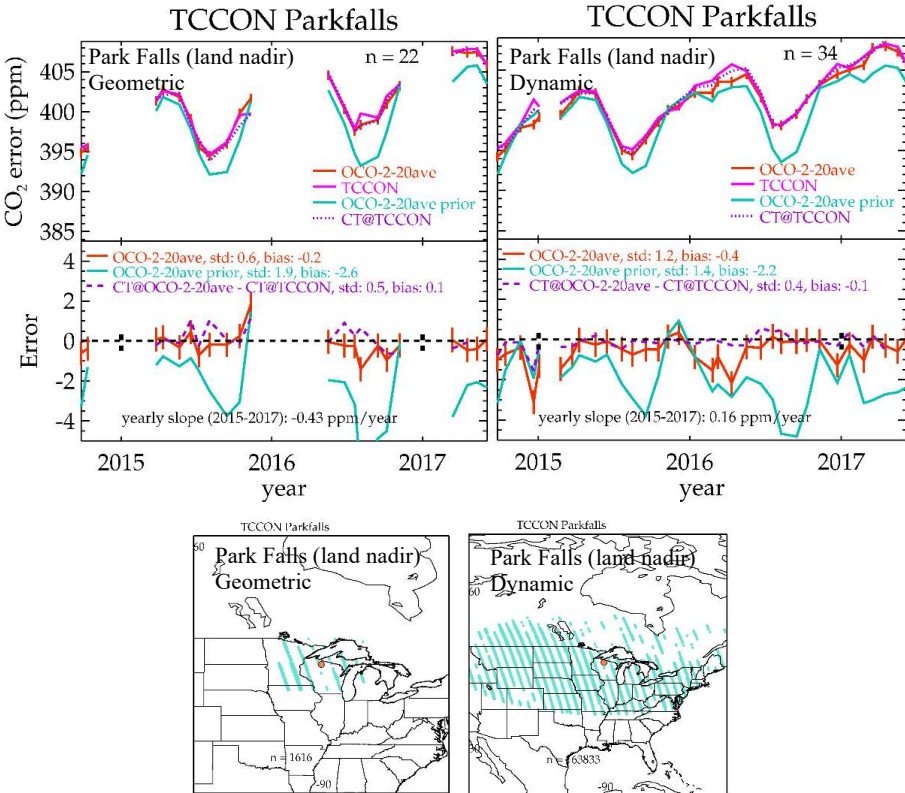

**Figure 4: Geometric and Dynamic coincidence criteria for OCO-2 versus TCCON at Park Falls. Same descriptions as Fig. 2. For v8 geometric coincidence, data is missing from November through April.**






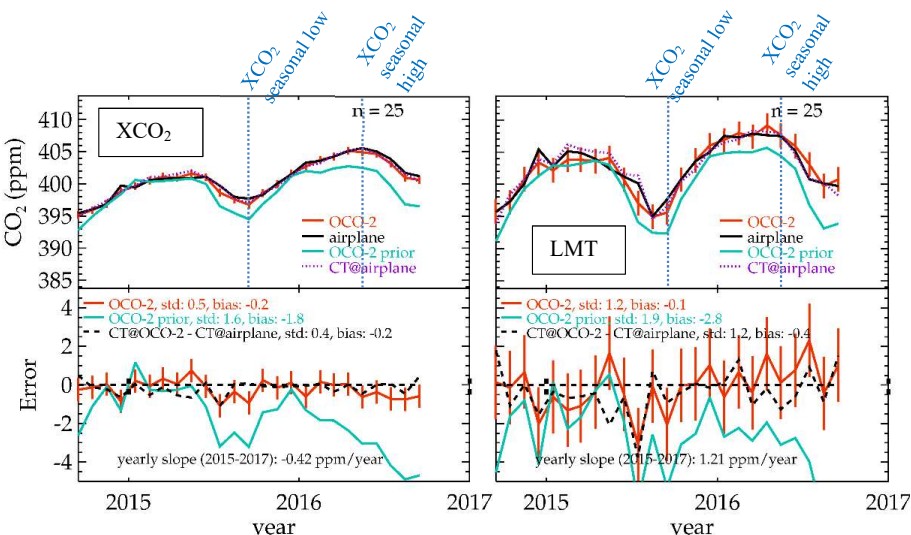

Figure 5: Monthly averages of OCO-2 XCO$_2$ and lower partial column (LMT) versus aircraft. The OCO-2 value is red, the prior in blue/green, and airplane in black. On the lower plots, differences are shown between OCO-2 (read) and the prior (blue/green) and aircraft. The lower plots show the differences between CarbonTracker co-located with OCO-2 minus CarbonTracker co-located with aircraft (black dashed lines).

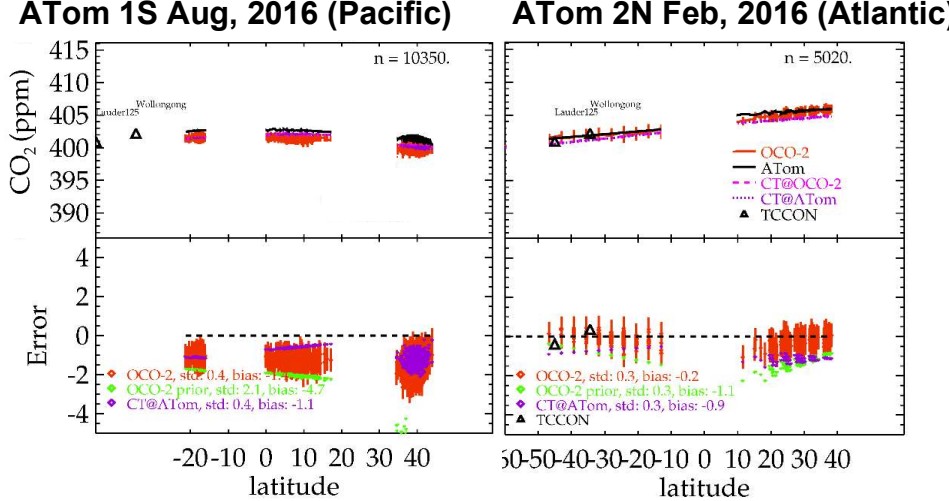


**Figure 6: OCO-2 ocean glint versus ATom for campaigns "1S" and "2N". OCO-2 (averaged over 10 adjacent observations) is shown in red, ATom CO2 in black. CarbonTracker co-located with to OCO-2 or ATom are shown in pink or purple, respectively. The lower plots show differences versus ATom. The difference between the OCO-2 prior and ATom is shown in green. TCCON co-**
**located in latitude and time are shown as black triangles. The gaps are locations where no co-located observations were found.**



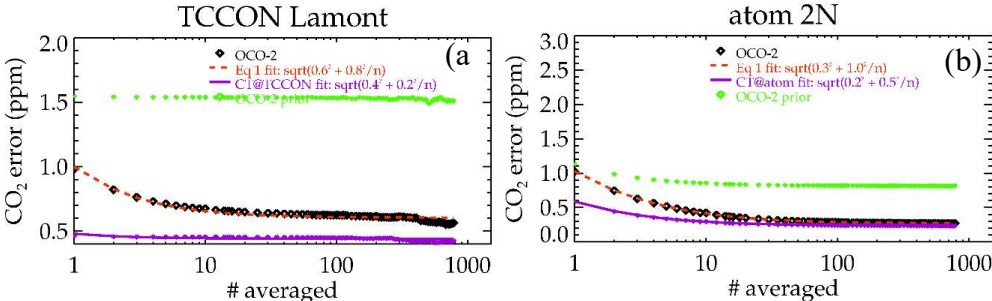

**Figure 7: Error reduction with averaging for (a) OCO-2 versus TCCON at Lamont (land nadir) for geometric coincidences, (b) OCO-2 versus ATom campaign 2N for ocean glint.**






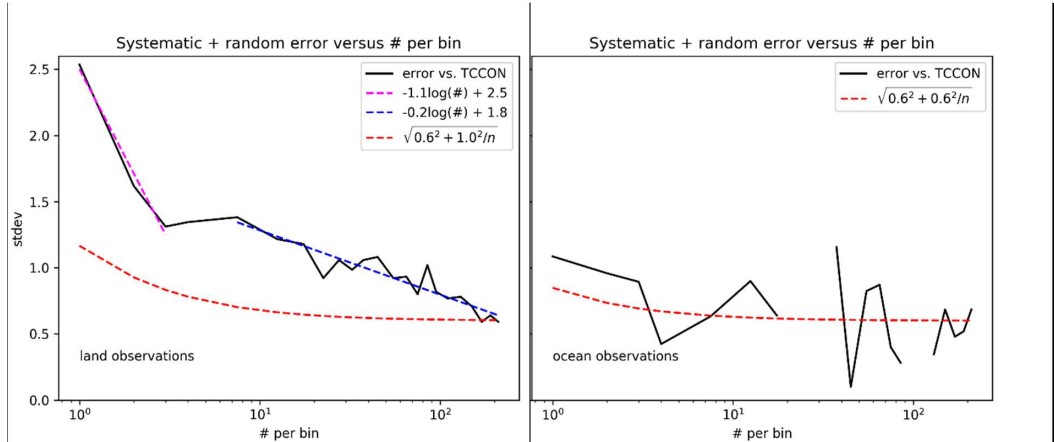

**Figure 8: Total error (systematic plus random) for the Baker-averaged product versus the number of observations per average for land (left) and ocean (right). The red dashed line shows the expected error from Table 2 and Eq. 3. The purple and blue dashed lines show fits for different regimes of $n$ for land. This shows that sparser data has higher error.**



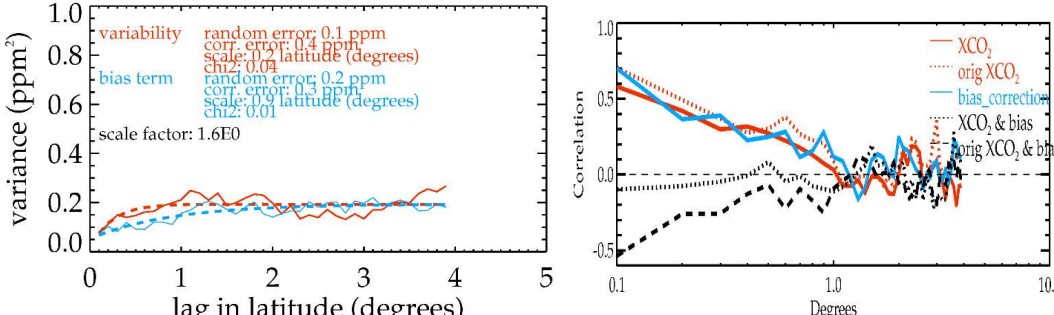

**Figure 9: (Left) Variogram of XCO₂ variability for remote ocean sites (left). For comparison the variogram of the bias correction term (blue) is also shown. (Right) correlation of XCO₂ variations (original, dotted; bias corrected solid), variations in the bias correction term (blue) and cross-correlations between the bias correction term and original XCO₂ (dashed) and bias corrected XCO₂ (dotted).**






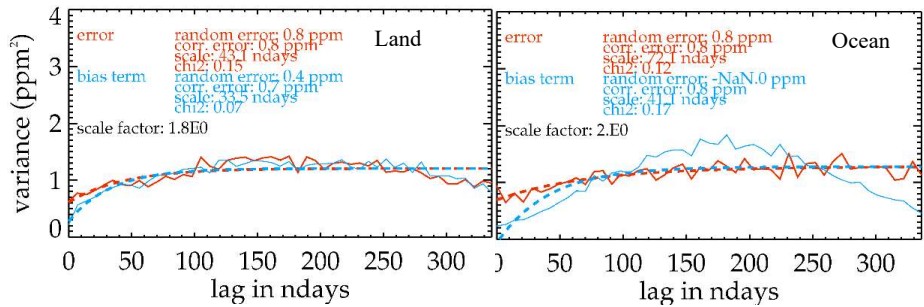

**Figure 10: Variogram of XCO₂ error (OCO-2 minus TCCON) in time for land (left) and ocean glint (right).**