# Peer review of "Characterization of OCO-2 and ACOS-GOSAT biases and errors for CO2 flux estimates"

_Atmospheric Measurement Techniques, 2019_

## Referee Comment (RC1) · Anonymous Referee #2 · 2 Dec 2019

This manuscript is, as far as I can tell, identical to the version submitted for access review. Therefore, this review is an updated and extended version of my access review.

In this manuscript, OCO-2 and ACOS-GOSAT CO2 products are validated with independent datasets. The focus of the manuscript is on OCO-2 XCO2 data. Error characterization of the satellite products includes random and systematic errors, correlation scales, and the errors of averaged products for OCO-2. The authors provide a rough estimate for the impact of OCO-2 XCO2 biases on fluxes on the scale of Transcom-3 regions. In addition to the results on XCO2, an OCO-2 LMT product is developed following the methodology of Kulawik et al., 2017, and uncertainties of both ACOS-GOSAT and OCO-2 LMT products are assessed.

The manuscript is clearly within the scope of AMT since it presents advances in green-

house gas remote sensing products. The authors mostly rely on methods and validation data sets that were employed/developed/presented in previous peer-reviewed publications. Thus, the manuscript has the potential to make a relevant contribution to satellite-retrieved atmospheric CO2. However, the validated OCO-2 XCO2 dataset is not up to date, and I have some concerns about some of the analyses and take-home messages. In addition, parts of the manuscript are not structured well and some of the methods are not explained clearly or in enough detail, which makes the manuscript hard to follow. Overall, I support publication of this work in AMT, but I have many comments and questions that need to be addressed.

GENERAL COMMENTS

SCIENTIFIC SIGNIFICANCE (GENERAL) Both ACOS-GOSAT and OCO-2 XCO2 products are continuously in development. Naturally, validation papers have been published before, but for the data versions presented here not to the extent done here. The parts on ACOS-GOSAT are in parts updates of previous work with a newer retrieval version (7.3 vs 3.5 in Kulawik et al. 2016, 2017). The OCO-2 evaluations are complementary and/or updates to previous work on previous versions of OCO-2 data (v8, while v7 was analyzed in Worden et al. 2017, Wunch et al. 2017) and on this data version (O'Dell et al., 2018).

Assessments like the one presented here are valuable for the CO2 inverse modeling community, in order to adequately account for observation errors in retrieving CO2 fluxes and their uncertainties. In this context, the characterization of error correlations is particularly commendable, since the inverse modeling community needs to improve in accounting for them. Characterization of errors of averaged products is likewise useful, because, as the authors state, this is what will be used by modelers.

However, the evaluated OCO-2 retrieval version is outdated. In my opinion, this impedes the usefulness of the manuscript, since data users will in most cases use the latest version. This issue is addressed in more detail in the major comments below.
To highlight the considerations concerning the OCO-2 data version, I rate the scientific significance of the manuscript "Fair".

SCIENTIFIC QUALITY Validation of CO2 remote sensing data by comparison to independent data is a well-established concept, and many of the methods the authors use are sound and explained sufficiently well. No dataset comes without biases, though, so differences between datasets will partly be due to the validation dataset and also validation method. The authors try to account for this by subtracting a validation error from the differences between validation and validated datasets, to isolate the contribution of the validated data. In several cases, this subtraction appears overly optimistic for OCO-2 / ACOS-GOSAT to me, i.e. some validation error components might be upper bounds or overestimated, which means that OCO-2 / ACOS-GOSAT errors could be underestimated. Since the validation-error corrected uncertainties are presented in the abstract and conclusions, they are the take-home message of the paper. Therefore, care must be taken that they are not underestimated, or at least it must be stated clearly that upper bounds of validation errors were subtracted. Other authors exercised more caution here (e.g. Wunch et al. 2017), and this manuscript would benefit from doing so as well.

The scope of the manuscript should be more clearly put in perspective with respect to previous and ongoing efforts of OCO-2 and ACOS-GOSAT algorithm development and product validation. For example, as mentioned below in my comments on the OCO-2 data version, Kiel et al. 2019 has to be cited, and it should be mentioned that O'Dell et al. 2018 already presented some validation of OCO-2 v8. Also, some analyses were only performed for OCO-2 – perhaps because similar analyses were done on ACOS-GOSAT in other papers?

In several cases, I couldn't match numbers given in the text to those in (referenced) tables or other sections (see specific comments).

I'm optimistic that the authors are able to clear up these points and therefore rate the

scientific quality as "Good".

SCIENTIFIC PRESENTATION The goals of the study are clearly laid out in the introduction, and abstract and conclusion sections contain adequate summaries of the results. Still, several aspects of the presentation have to be improved. In some cases, important details on methods are missing, and some parts of Sections 3 and 4 could be structured much better (see comments below). I also agree with the other access review that the general figure quality is not good (compressed, some labels crossed by data or axes).

The above points are in principle not hard to fix. Therefore, I rate the scientific presentation as "Good".

MAJOR COMMENTS

OCO-2 DATA VERSION The evaluated OCO-2 retrieval version - v8 - was superseded by v9 in 2018 (the user guide is dated October 2018). OCO-2 XCO2 v9 was documented in Kiel et al., 2019 and in further documentation available at https://doi.org/10.5067/W8QGIYNKS3JC. The authors do not mention this update or take into account known biases of the v8 product that were corrected for in v9. Since the main purpose of the manuscript is error characterization, the results related to OCO-2 XCO2 may be of limited use due to the already-available update.

Here, I attempt to assess the relevance of the update for the conclusions of the manuscript. According to Kiel et al. 2019, differences between v8 and v9 are on the order of up to +/- 0.5 ppm on 2x2deg scale globally, but mostly below +/- 0.4 ppm, and with a mean bias of 0.15 ppm (Kiel et al., 2019, Fig. 14). In the manuscript here, systematic OCO-2 XCO2 v8 errors are reported to be on the order ∼0.6 ppm. This is based on comparisons with independent datasets, with co-location criteria including areas that are much larger than 2x2 degree, but where v9-v8 differences might in some cases be coherent (Kiel et al. 2019, Fig. 14). Random errors might be affected by the v9 update as well, since it corrects for an error correlated with terrain slopes, which

can vary on smaller scales than those encompassed by the co-location criteria used here. In addition, the OCO-2 bias correction term uses dP (difference between prior and retrieved surface pressure), which is directly affected by the update to v9. Other conclusions than that of systematic errors affected less or not at all, including temporal error correlations, and of course everything related to ACOS-GOSAT v7.3. In summary, the v9-v8 differences are probably mostly smaller than the v8 errors reported here, but I wouldn't rule out significant differences in certain areas. Of course, this assessment is far from conclusive.

Further considerations about the data version are that some limited v8 validation (i.e., comparison to TCCON) was already presented in O'Dell et al., 2018, and v9 is already being used in scientific studies (Zheng et al., 2019, Reuter et al., 2019).

In my access review, I suggested considering an update to OCO-2 v9. This would extend the period for which (some of the) conclusions of this paper stay relevant. An argument against such an update is that the validated ACOS-GOSAT product (v7.3) largely corresponds to the OCO-2 v8 algorithm. For the sake of consistency, it makes sense to validate these products together. The authors should comment on this suggestion. At the very least, the authors should reference OCO-2 v9 and add statements to the manuscript about to what extent they expect the conclusions they obtained for v8 to be valid/will change in v9.

ESTIMATES OF VALIDATION ERRORS (SECTION 3.4) As stated above, I have the impression that several validation errors represent upper bounds, which is also stated in l 355, or might even be overestimated in some cases, but this is not reflected in the way ACOS-GOSAT/OCO-2 errors are reported in the abstract and conclusions. The authors need to acknowledge that part of this subtracted error might be attributable to the tested datasets. A concrete example that indicates that validation errors could be overestimated in this study is in ll 474-476, where an OCO-2 error estimate is smaller than the validation error. In the following, I comment on individual validation error estimates as described in Sect. 3.4.

Co-location error ll 359-361: Please add a paragraph or cross-references to paragraphs where the computation of biases, correlated and random co-location errors is explained.

TCCON error The 0.4 ppm used here are the TCCON site-to-site bias variability. Conceptually this is OK, but attributing the full variability solely to the validation dataset is optimistic because it assumes that the satellites don't suffer from the same biases. I doubt this because the satellite data are bias-corrected based on TCCON data. Wunch 2017 were more cautious and stated that differences within this error (0.4 ppm) _may_ be attributable to TCCON. The TCCON error should be treated in the same way here.

Averaging kernel error I don't understand the description of the averaging kernel difference error in ll 369ff. Section 2.1.3 states that TCCON is compared to OCO-2 by applying the averaging kernel of the latter to the retrieved profile of the former. This already accounts for the different averaging kernels, to the extent possible. There is a remaining smoothing error. The examples given in ll 369ff (0.58 ppm -> 0.5 ppm and 0.82 ppm -> 0.72 ppm) show the error reduction achieved by using the correct averaging kernel as opposed to ignoring averaging kernel differences. In the notation of Rodgers and Connor 2003 ("RC03"), perhaps this describes the difference between $c_1-c_2$ (RC03, Eq. 23) and $c_1-c_{12}$ (RC03, Eq. 26)? This wouldn't be exactly be the smoothing error that remains after applying the averaging kernels as described in Sect. 2.1.3, which is $c_1-c_{12}$ (RC03, Eq. 26). Please clarify what the final estimate of 0.1 ppm averaging kernel error represents, e.g. by referencing RC03 or using its notation.

Aircraft profile sampling and extension to surface It is not clear to me how aircraft profiles were extended to the surface. Explanations are scattered through several sections, and I don't know which method was applied to which dataset/for which analyses it was used: in ll 418ff, a profile extension using AirCore data is described, but this only applies to one site (SGP). Section 2 states "the profile is extended to the surface using the lowest observation" without further explanation. Section 3.1 briefly mentions:

"The aircraft profile is extended with CarbonTracker2017" about LMT. Please bundle the explanations on aircraft profile extension in one section or paragraph somewhere. In cases where CarbonTracker is used to extend profiles to the surface, aircraft error could be overestimated, because CarbonTracker, as a global model, likely can't reproduce the variability of $CO_2$ close to the surface at the scale of observations. This would lead to an underestimation of satellite error. As stated above, this would need to be accounted for in the manuscript.

SECTION 4.2 In Sect. 4.2, the authors state that they "cannot compare [LMT] versus TCCON because TCCON is a total column measurement, so LMT is compared to aircraft and remote surface site observations". However, LMT is also a very different product than surface data, sampling part of the column vs a point close to the surface. This difference needs to be accounted for in some way in the manuscript. The authors could add a paragraph on why differences between these sampling strategies do not affect their results, or, more convincingly, could assess this assumption based on the atmospheric $CO_2$ models already used for similar purposes in the study.

SECTION 5.2.1 In this section, error correlations of XCO2 by OCO-2 are inferred by assuming that the true XCO2 is constant over the latitude range in a remote ocean area. I think the box used here could be too large for this assumption. It covers 30° latitude and latitudinal XCO2 variations could be affected by interhemispheric mixing of the NH seasonal $CO_2$ flux and synoptic variability. Aren't usually longitudinal variations in remote ocean locations used for this kind of analysis? I doubt that the assumption of constant XCO2 on each day throughout the box holds to sub-ppm level. Worden et al., 2017 used much smaller areas for a similar analysis. In my opinion, the authors have to discuss the validity of their assumption of absent natural variability, and/or the influence of violation on the results. This could be done similarly to my suggestion on Sect. 4.2 above, i.e. by repeating the analysis with a sampled model. This doesn't have the resolution required to detect the correlation scale of 0.2 degrees latitude, but could shed light on what happens at larger lags. In particular, I wonder why this method

didn't pick up the larger correlation length-scale of 5-10 degrees found in Sect. 5.2.2. The authors should also provide uncertainties of the parameters estimated based on the variograms in Fig. 9. Especially: Can the short correlation length scale of XCO2 (only twice the resolution of binned data) really be retrieved, or is the uncertainty too high?

SECTION 5.3.1 (Simplistic calculations of flux estimate uncertainty) In Sect. 5.3.1, the authors estimate flux biases that result from XCO2 biases with a simple mass balance approach. While this is a plausible concept, it does not reflect how regional and global flux estimation works, because it ignores atmospheric transport and that models evaluate gradients, which the authors acknowledge implicitly, transport model spin up, and spatial and temporal correlations that are typically imposed on fluxes to regularize the solution. Consequently, the results contradict the results in 5.3.2, where flux errors on similar spatial scales are estimated to be much larger based on an inverse model. Therefore, the results in this section are not useful. On the contrary, they suggest that XCO2 errors have less impact on flux estimation than they actually have. Since the section is isolated from the rest of the manuscript, I suggest to remove it. Also, I haven't found the correspondence 1 ppm <=> 2.1 PgC in Baker et al. (2006) in a quick search.

SECTION 5.3.2 Section 5.3.2 provides estimates for the impact of XCO2 biases. The authors should include details on the inverse model setup, because this would be useful to evaluate the results. This includes which inverse model and transport model were used, which XCO2 product was assimilated, and prior flux and XCO2 uncertainties. Why was the section based on the OCO-2 v7 bias correction term, while previous sections characterized the v8 bias correction term? The authors cite similarity in magnitude and correlations to their XCO2 bias for the choice to assimilate the bias term, but in Figures 9 and 10, the bias terms were scaled to match the XCO2 errors – was this taken into account? According to ll 636-638, the mapping of flux biases to individual regions is meaningless. For this reason, Table 8 is misleading and only aggregate

results like averages and ranges should be given. What's the reference for the flux magnitudes of 0.2 PgC/yr for land and 0.5 PgC/yr for ocean Transcom regions? A similar analysis of the LMT product could add value to the manuscript, since this could strengthen the case for its usefulness in inverse modeling studies.

APPENDIX A I would like to see more details on how the variables for the bias terms were selected. Were there other plausible choices? What about collinearity between covariates? Did the authors estimate the impact of choosing different plausible sets of covariates (if there were any) on the spatiotemporal distribution of the bias term? This might have impacts on errors in retrieved fluxes.

LANGUAGE Please harmonize the use of present / past tense throughout the manuscript (e.g. l 653 vs l 655)

STRUCTURE

Section 3 Section 3 is supposedly about methods, and specifically, "Section 3.1 describes the types and methodology used to estimate systematic and random errors" (ll 281f). However, both Sect. 3.1 and 3.2 contain results. These should be moved to another section or the scope of Sect. 3 should be adjusted.

Section 3.4 needs to be restructured. First, a list with bullet points is given, then paragraphs with additional information follow for some bullet points. However, some bullet points are already very long. All this could be fixed by using subsections instead of bullet points.

Section 4 Section 4.2 has almost the same caption as Sect. 4. Section 4.1 is very short and falls into the scope of Sect. 4.2.

SPECIFIC COMMENTS

l 57: Add average land bias reduction in ppm

l 85: two -> three

ll 94ff: The manuscript goals as outlined here mostly focus on OCO-2 XCO2 data. It is unclear to me why some analyses were only performed for OCO-2 and not for ACOS-GOSAT, which makes parts of the manuscript a bit patchy. If there are reasons behind this choice, this would be a good spot to put them.

ll 109ff: Introduce labels of ATOM validation datasets (1S, 2N, Atlantic, Pacific)

l 112: Reference to Frankenberg et al., 2016 missing in bibliography

l 152: This explanation is confusing (e.g. the term "original observation"). State somewhere that one retrieval is simulated with the profile obtained by the other, perhaps use equations.

ll 240f: You could mention that other authors compute observation errors differently. E.g. Crowell et al. 2019 use, among other estimates, the maximum of 1) and 2) instead of the sum. One might argue you are double-counting errors, though I acknowledge there is no one correct way, since no uncertainty estimate is accurate.

ll 241-243: I don't understand "The means of the observation uncertainty. . . were used to represent biases". I'm not familiar with the Liu average product, but isn't the observation uncertainty reported in the OCO-2 lite files the posterior uncertainty of the Bayesian retrieval, which represents random errors? I don't see how this could be a useful estimate for a bias.

l 291: The term "dynamic small" is only mentioned here. It sounds like two sets of criteria were used for different datasets or comparisons. Please clarify which criterion was used in which analysis, perhaps change "dynamic" to "dynamic small" / "dynamic large" throughout the manuscript.

l 315: "The co-location flags a large co-location error in mid-2015" - sentence unclear.

l 336: "feeds into" – please be specific here.

l 355: "These errors quantified result from" - ? (probably delete "quantified" or "These"

-> "The"?)

l 363: 4 -> Four

l 372: look -> looking (also, better split this very long sentence)

l 394: Remove one sentence fragment.

l 395: The AirCore dataset is not introduced nor is a reference for it given. Please add an introduction to the AirCore dataset in Sect. 2.

l 453: Please clarify what these numbers are. I think it should be those in Table B1, but the numbers don't match. Text: Manaus -1.6 ppm, Hefei -1.2 ppm. Table B1: Manaus -1.2 ppm, Hefei: -0.8 ppm.

ll 457ff: Are these the "corrected systematic errors" from Tables 2, …? Please clarify. I'm not sure because CarbonTracker doesn't have 0.4 ppm (l 463) but 0.4-0.6 ppm errors according to Table 3. Also: given that uncertainties in the validation data sets are a key point in this manuscript, model-data mismatch cannot be attributed unambiguously to errors of the CarbonTracker model, which the wording here suggests.

l 465: Remove "larger"

ll 468, 476, 480, 724, maybe others: I suggest to refer to surface observations as "flask" observations. This would make the connection to their introduction in Sect. 2.1.4 easier.

ll 478f: Reference for this low bias of aircraft measurements? To which aircraft dataset does this refer, and does it make sense to assume it applies here as well?

l 506: Why does the Baker product (Fig. 8) have so much larger errors than "direct" averages (Fig. 7) for small numbers of observations? Aren't both datasets based on the same XCO2 product? Specifically, what uncertainty estimate should I use if I wanted to use data from an average product in a low throughput region for CO2 flux estimations?

ll 580ff (Section 5.2.2): Please provide a figure to illustrate the results of this section.

l 653: Averaged quantities -> averaged XCO2 products (?)

l 654: The 2.6 ppm were 2.5 ppm in Sect. 4.3 (?)

ll 657f: Does this statement refer to OCO-2?

ll 660f: The high errors in low-throughput data are a very important finding! Has this been reported or hinted at before? If so, please include references.

Table 1: Please add to the caption that this table is about OCO-2 only. The footnote could be moved to the caption.

Table 2: What's "non-sequential averaging"?

Table 7 is not referenced in the text.

Figures 2-4: Move the maps into additional figures or give them panel names

Figure 9: Mention in the caption that this is for OCO-2. Please also check if this information is missing in other figures.

Figure 9: Judging from the noise and other variograms presented here, the dip around lag 2°-3.5° looks significant. Out of curiosity, do you have an idea what causes it?

REFERENCES

Crowell, S., Baker, D., Schuh, A., Basu, S., Jacobson, A. R., Chevallier, F., Liu, J., Deng, F., Feng, L., McKain, K., Chatterjee, A., Miller, J. B., Stephens, B. B., Eldering, A., Crisp, D., Schimel, D., Nassar, R., O'Dell, C. W., Oda, T., Sweeney, C., Palmer, P. I. and Jones, D. B. A.: The 2015-2016 carbon cycle as seen from OCO-2 and the global in situ network, Atmos. Chem. Phys., 19(15), 9797–9831, doi:10.5194/acp-19-9797-2019, 2019.

Kiel, M., O'Dell, C. W., Fisher, B., Eldering, A., Nassar, R., MacDonald, C. G., and Wennberg, P. O.: How bias correction goes wrong: measurement of XCO2 affected

by erroneous surface pressure estimates, Atmos. Meas. Tech., 12, 2241–2259, https://doi.org/10.5194/amt-12-2241-2019, 2019.

Kulawik, S., Wunch, D., O'Dell, C., Frankenberg, C., Reuter, M., Oda, T., Chevallier, F., Sherlock, V., Buchwitz, M., Osterman, G., Miller, C. E., Wennberg, P. O., Griffith, D., Morino, I., Dubey, M. K., Deutscher, N. M., Notholt, J., Hase, F., Warneke, T., Sussmann, R., Robinson, J., Strong, K., Schneider, M., De MazieÌĂre, M., Shiomi, K., Feist, D. G., Iraci, L. T., and Wolf, J.: Consistent evaluation of ACOS-GOSAT, BESD-SCIAMACHY, CarbonTracker, and MACC through comparisons to TCCON, Atmos. Meas. Tech., 9, 683–709, https://doi.org/10.5194/amt-9-683-2016, 2016.

Kulawik, S. S., O'Dell, C., Payne, V. H., Kuai, L., Worden, H. M., Biraud, S. C., Sweeney, C., Stephens, B., Iraci, L. T., Yates, E. L., and Tanaka, T.: Lower-tropospheric CO2 from near-infrared ACOS-GOSAT observations, Atmos. Chem. Phys., 17, 5407–5438, https://doi.org/10.5194/acp-17-5407-2017, 2017.

O'Dell, C. W., Eldering, A., Wennberg, P. O., Crisp, D., Gunson, M. R., Fisher, B., Frankenberg, C., Kiel, M., Lindqvist, H., Mandrake, L., Merrelli, A., Natraj, V., Nelson, R. R., Osterman, G. B., Payne, V. H., Taylor, T. E., Wunch, D., Drouin, B. J., Oyafuso, F., Chang, A., McDuffie, J., Smyth, M., Baker, D. F., Basu, S., Chevallier, F., Crowell, S. M. R., Feng, L., Palmer, P. I., Dubey, M., GarciÌĄa, O. E., Griffith, D. W. T., Hase, F., Iraci, L. T., Kivi, R., Morino, I., Notholt, J., Ohyama, H., Petri, C., Roehl, C. M., Sha, M. K., Strong, K., Sussmann, R., Te, Y., Uchino, O., and Velazco, V. A.: Improved retrievals of carbon dioxide from Orbiting Carbon Observatory-2 with the version 8 ACOS algorithm, Atmos. Meas. Tech., 11, 6539–6576, https://doi.org/10.5194/amt-11-6539-2018, 2018.

Reuter, M., Buchwitz, M., Schneising, O., Krautwurst, S., O'Dell, C. W., Richter, A., Bovensmann, H., and Burrows, J. P.: Towards monitoring localized CO2 emissions from space: co-located regional CO2 and NO2 enhancements observed by the OCO-2 and S5P satellites, Atmos. Chem. Phys., 19, 9371–9383, https://doi.org/10.5194/acp-19-9371-2019, 2019.

Rodgers, C. D. and Connor, B. J.: Intercomparison of remote sounding instruments, J. Geophys. Res. D Atmos., 108(3), doi:10.1029/2002JD002299, 2003.

Worden, J. R., Doran, G., Kulawik, S., Eldering, A., Crisp, D., Frankenberg, C., O'Dell, C., and Bowman, K.: Evaluation and attribution of OCO-2 XCO2 uncertainties, Atmos. Meas. Tech., 10, 2759-2771, https://doi.org/10.5194/amt-10-2759-2017, 2017.

Wunch, D., Wennberg, P. O., Osterman, G., Fisher, B., Naylor, B., Roehl, C. M., O'Dell, C., Mandrake, L., Viatte, C., Kiel, M., Griffith, D. W. T., Deutscher, N. M., Velazco, V. A., Notholt, J., Warneke, T., Petri, C., De Maziere, M., Sha, M. K., Sussmann, R., Rettinger, M., Pollard, D., Robinson, J., Morino, I., Uchino, O., Hase, F., Blumenstock, T., Feist, D. G., Arnold, S. G., Strong, K., Mendonca, J., Kivi, R., Heikkinen, P., Iraci, L., Podolske, J., Hillyard, P. W., Kawakami, S., Dubey, M. K., Parker, H. A., Sepulveda, E., GarciÌĄa, O. E., Te, Y., Jeseck, P., Gunson, M. R., Crisp, D., and Eldering, A.: Comparisons of the Orbiting Carbon Observatory-2 (OCO-2) XCO2 measurements with TCCON, Atmos. Meas. Tech., 10, 2209–2238, https://doi.org/10.5194/amt-10-2209-2017, 2017.

Zheng, T., Nassar, R., and Baxter, M.: Estimating power plant CO2 emission using OCO-2 XCO2 and high resolution WRF-Chem simulations. Environmental Research Letters, 14(8), 085001, https://doi.org/10.1088/1748-9326/ab25ae, 2019.

---

## Referee Comment (RC2) · Anonymous Referee #1 · 22 Jan 2020

General Comments:

The manuscript describes an evaluation of biases in remotely sensed CO2 concentration from GOSAT and OCO-2 using the ACOS retrieval algorithm. It is extremely critical to understand such biases in order to make full use of these remote sensing datasets within flux inversion frameworks. In practice it is very difficult to obtain this information because comparison datasets are much sparser, sample different atmospheric volumes, and have their own biases relative to the unknown truth. As such, I think the focus of the paper is very timely and useful to the research community. As far as I am aware, this is a novel study because it brings together multiple comparison datasets (TCCON, aircraft, and surface flask data) and also attempts to characterize errors and biases in some of the comparison methods themselves, such as sampling

mismatches.

However, the paper itself has numerous problems in presentation and some potential methodological problems. The major specific problems and my suggestions are listed below, grouped into respective sections.

Methodology:

In general, I question whether all the quoted uncertainties should have been rounded to 0.1 ppm. Since most of the uncertainties are < 1.0 ppm, this means most numbers are rounded to a single significant figure. This causes problems in interpretation of many of the results - for example, just the first set of numbers in Table 1, we have the total co-location error for Geometric (CT2017) listed as 0.4 (0.3, 0.2). I think the value of 0.4 is the quadrature sum of 0.3 and 0.2; if we had two significant figures this quadrature sum could actually range from sqrt(0.34**2 + 0.24**2) = 0.42 to sqrt(0.26**2 + 0.16**2) = 0.31. This problem is worse later when two uncertainties are subtracted (Table 2). The authors should report an additional figure, or explain the justification for rounding to one significant figure.

Line 315 - "The LMT reaches the maxima and minima at least one month before XCO2" I do not think this statement is not supported by figure 5. The error bars on the time series are very large, so the claimed time offset does not appear to be statistically significant. More sophisticated analysis needs to be performed to support this claim. In addition, I would argue this aspect of the data is unrelated to the focus of the paper, so should be omitted.

Section 4.2. Line 482 - To assess the relative variability of LMT versus XCO2, the authors are "Looking at a few random days" - this is not sufficient analysis, particularly since the data underlying the manuscript is much more extensive. Can you just compare the overall daily standard deviations, averaged over the multiple years of available data? Otherwise the reader is left the impression these 3 days are selected for some omitted reason.

Section 5.3.1. - These simplistic calculations are too simple in my opinion. There is some dimensional mismatch that is not explained: why is a concentration anomaly - the regional bias - directly converted to a flux? The Baker 2006 numbers are in PgC/year, while the anomalies are ppm. I do not see any need for this section at all, since the following section (5.3.2) is a much more realistic representation of how the concentration anomalies would actually impact flux estimates. I would also add detail to section 5.3.2, there are some lacking details. How is the bias correction term assimilated, exactly? How do you explain 'no overall bias resulting from the bias [correction term] assimilation', when the mean of the bias correction term is nonzero?

Presentation:

I would recommend the paper be reviewed by a co-author or colleague, focusing on improving the clarity of the language. In particular, I feel there is a lack of consistency of what is precisely meant with terms like "bias", "error", "correlation", which makes it very difficult for the reader to understand the analysis.

Somewhere in Section 1 or 3 (I am not sure the best place), there need to be equations spelling out how the different error terms are applied to either the satellite data, the validation data, or the differences. There must be some assumed underlying statistical model here; explaining that clearly would make the manuscript much more understandable. This would also make the different terms less ambiguous.

Section 3.4 needs to be reorganized. There is a bulleted list of different uncertainty aspects of the validation process, but each bulleted item has a long description. The last bullet however, just says "this is described in more detail below", which is then just the rest of section 3.4. This could be split into subsections, or just make one short abbreviated list followed by separate paragraphs describing each aspect. The section describing the averaging kernel error is very unclear.

Section 5.2, there is no specific definition of what quantity is being computed, I am not sure what "the square of the XCO2 error versus time" means - I think this is similar to

the Torres et al 2019 paper, but in that paper they clearly define the statistic they are using first (the semivariance, equation 8 in the paper), and then the analytic model they use (equation 9). What statistic is being used here?

Later in Section 5.2.1, the section describing correlations of bias correction terms versus the XCO2 is not clear and needs more explanation. In addition, I do not find this part of the manuscript is well aligned to the main focus of the manuscript (characterization of biases relative to validation data), and I would recommend removing it.

Section 5.2.3, this section also needs an unambiguous description of the statistic being used. "Time correlation of the error (OCO-2 minus TCCON) is used to calculate correlation of error versus time." - this is not meaningful.

Another common issue in the presentation, is there are many unneeded parenthetical statements, or unrelated comments. Some examples: Line 155: "It would also make more sense to use a more relaxed co-location ... but we found sufficient co-locations within +- 1 hour" This entire sentence could be omitted, there is no reason to discuss some processing aspect that was not needed. Line 295: this statement about other co-location techniques is unrelated to what was actually used, it should be removed or moved to the discussion section at the end. Line 392: "... described below (although it is possible that different models or schemes have biases in the same direction)." This is unclear and seems unrelated to the rest of the paragraph.

Figures: these are all very low quality, difficult to read, and not publication quality. Particularly figure 1, where many of the labels are unreadable. If possible, all figures should be recreated at a higher resolution and not compressed (specifically, use PNG not JPG).

Figure 2, and other similar plots: in the lower panels that show difference plots, there is appears to be a dashed line along y=0 that looks the same as the CarbonTracker derived difference. Remove the y=0 lines (otherwise I cannot tell which is which) or replot the y=0 as a dotted or thinner dashed. What do the vertical marks at 2015 and

2017 on the y=0 line signify?

Figure 6 - I cannot tell what this represents, are these boxplots, or just scatter plots where all the points are piled on top of each other? Please use a more visually intuitive plot here.

Figure 8 - the linear fits on the left side are not very meaningful. It appears that the fit for the small bin numbers is a linear fit to 3 data points. This is not a robust result. If the authors want to claim a robust difference in slope between the two scale ranges, this needs a more sophisticated analysis, with an assessment of statistical significance between the two slopes.

Figures 9, 10 - Legends need to be plotted with better spacing. Why does the bias term fit in figure 10 have a -NaN value? What are the chi2 values listed in the legend? If these are the typical reduced chiˆ2 values, why are they all well below 1?

Minor issues:

Line 57 "systematic errors by a factor of 2 for land observations and improves errors by $\sim$0.2 ppm for ocean." Describe the magnitude of the errors either multiplicatively or additively for both datasets, instead of mixing the types. The reader cannot assess which data has higher systematic errors with this description.

Line 70: The description of OCO-2 observation modes is not accurate. There are three modes: Nadir, Glint, Target; the data in the "land glint" and "ocean glint" datasets is collected in the same mode. Later in Line 74 the phrase "standard modes" is referenced but not described.

Line 81: Need a reference for the CO2 variations. Is this XCO2 or CO2 concentrations anywhere in the column?

Line 95: the bulleted list is not clear: What "effects" of bias correction are being characterized? this is too vague. The last bullet does not agree with the lead in sentence fragment - a clearer statement might be "the magnitude of false surface fluxes induced

by regional biases assimilated in an inversion model"

Line 110: there needs to be a DOI, or at least a URL, for where this data file was obtained.

Line 142: here "The TCCON averaging kernel is applied" but then in line 145 "the OCO-2 averaging kernel applied" - which is it?

Line 160 - 165, what is 'Lauder125' and 'Tsukuba125' ? are the "125" suffixes meaningful or are these typos?

Line 218 - Please define the "predicted total error from the v8 lite product"? Is this the xco2 uncertainty from the level 2 algorithm?

Line 220 - what uncertainty is assigned to the average of the 1-second averages?

Line 221 - now this is "85 observations per bin" - but this is 85 observations per 10-second average, correct? This section would be clearer without using the word "bin" for each stage in the averaging process, or perhaps call them "10-second bins" or "1-second bins" as appropriate

Line 244 "The field "xco2_std+uncertainty" is the predicted error for the averaged product." Is this is referring to a specific field inside the v8 Liu dataset? Is this relevant? There is no publicly available dataset in this case, so I am not sure what use this information is to the reader.

Line 288 - Keppel-Aleks 2012 used the potential temperature at 700 hPa, is that what was used here?

Line 478 - "However, recently aircraft measurements were found to have a low bias on the order of -0.6 ppm for LMT ..." reference needed.

Line 670 "The LMT product is most useful when the 2 partial 670 columns behave differently" - This sentence is unclear and vague, what is meant by "behave differently"?